# Training-free Counterfactual Explanation for Temporal Graph Model Inference

**Mingjian Lu[1], Haolai Che[1], Yangxin Fan[1], Qu Liu[2], Fei Shao[1],**
**Tingjian Ge[2], Xusheng Xiao[3], Yinghui Wu[1]**

[1]Case Western Reserve University
[2]University of Massachusetts Lowell
[3]Arizona State University

## Abstract

Temporal graph neural networks (TGNN) extends graph neural networks (GNNs) and have demonstrated promising performance for dynamic and spatiotemporal network analysis. However, interpreting TGNN over dynamic graphs remains far less explored. This paper introduces TEMporal Graph eXplainer (TemGX), a training-free, model-agnostic, and query-able framework to help users interpret and understand TGNN-based graph analysis. TemGX discovers temporal subgraphs and their evolution that are responsible for inference results of interest, in terms of temporal counterfactual analysis. We introduce a class of explainability measures that integrate spatial-temporal influence and time decay model, to capture temporal influence contextualized by sliding windows. We formulate the explanation task as a constrained optimization problem, and present fast algorithms to discover explanations with guarantees on their temporal explainability. Our experimental study verifies the effectiveness and efficiency of TemGX for TGNN explanation, compared with state-of-the-art explainers. We also showcase how TemGX supports temporal queries for interpretable dynamic network analysis.

## 1 Introduction

Temporal graphs provide a powerful abstraction to represent complex dynamic systems, where nodes specify evolving entities, and edges denote temporal relationship. Temporal Graph Neural Networks (TGNNs) (Kazemi et al., 2020) model temporal graphs by incorporating temporal features, relations, and their dependencies, making them capable of predicting evolving interactions. TGNNs have demonstrated their effectiveness in various tasks, such as link prediction and node classification, in various applications *e.g.,* traffic analysis, transaction networks. In a nutshell, a TGNN $\mathcal{M}$ takes as input a temporal graph (a sequence of graph snapshots) $\mathcal{G}_t = \{G^1, \ldots G^{t-1}\}$, where each snapshot $G^i$ is represented by a pair $(X_i, A_i)$ of a node feature matrix $X_i$ and an adjacency matrix $A_i$ at time $i$ $(i \in [1, t])$. It is trained to predict the representation $(X_t, A_t)$ of a next graph $G^t$, which can then be transformed to task-specific output, such as class labels for node classification, features for regression analysis, or new links (events) for link prediction.

Despite their powerful predictive capabilities, TGNNs are often considered lack of interpretability. The decision-making process of TGNNs is deeply embedded within multiple layers of nonlinear transformations and complex temporal mechanisms, which result in opaque predictions (Xia et al., 2022). Such "black-box" behavior raises significant concerns in applications requiring transparent, trustworthy, and accountable temporal graph analysis, such as financial transactions or health-related predictions. Consider the following example.

**Example 1:** In Bitcoin blockchain transactions, money launderers employ a variety of techniques to conceal illicit cryptocurrency activities and evade detection by law enforcement agencies and AI-based monitoring systems (Bellei et al., 2024; Cheng et al., 2023). As shown in Figure 1, an input temporal graph $\mathcal{G}$ includes account IP addresses and Bitcoin transactions among the accounts at different timestamps.

A temporal edge $(v_1, v_2, '2')$ in $G$ represents a transaction committed from IP address $v_1$ to address $v_2$ at timestamp 2. Each IP address (a temporal node) is associated with a set of transaction features, *e.g.,* the number of blockchain transactions, the number of transacted bitcoins, and the amount of fees in bitcoins. There are two illicit IP addresses: $v_1$ and $v_t$. They utilize money laundering patterns known as *Peel Chain* (Bellei et al., 2024) ($P_1$) and *Spindle* (Cheng et al., 2023) ($P_2$) to obscure their illegal transactions. The perpetrators take *Spindle* strategy to generate multiple shadow addresses, and transfer illicit assets along lengthy paths that converge at a specific destination. Each path typically involves small amounts to avoid detection. *Peel Chain* launders cryptocurrency through a series of small transactions that are 'peeled' from the original address and sent to exchanges for conversion into fiat currency or other assets to minimize risks.

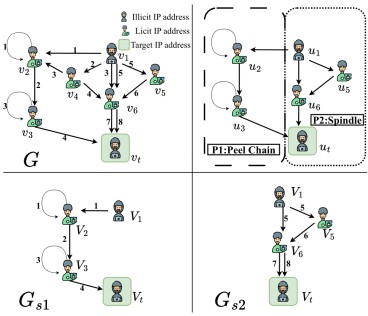

Figure 1: A temporal bitcoin transaction network $G$ that depicts money laundering activities (nodes: IP addresses; edges: transactions).

A TGNN-based classifier detected both accounts as "illicit" at timestamps 4 and 9, respectively. Law enforcement needs evidence to validate these results. A temporal subgraph $G_{s_1}$ explains "why" $v_t$ is "illicit" by containing historical transactions during period $[1, 4]$—if not captured in history, TGNN fails to detect $v_t$ as "illicit" (counterfactual explanations (Kuratomi et al., 2024; Lucic et al., 2022)).

Multiple such structures exist over time, requiring a "global" model to fit their distribution. This summary can validate known patterns like "Peel Chain" and reveal how strategies "evolved" over time, *e.g.,* $v_1$ has temporal dependency with $v_t$ via suspicious transactions, with dependencies evolving into new patterns during period $[5, 8]$, indicating switching from "Peel Chain" to "Spindle". □

The above example calls for effective method that can (1) generate "instance-level" explanations that expose activities as temporal subgraphs that are responsible for TGNN outputs in terms of counterfactual analysis; (2) summarize such structures over time as "global" statistical models, and (3) be conveniently queried to understand their evolving interactions. Several methods have been proposed for TGNN explanation *e.g.,* (Xia et al., 2022). These methods aim to trace the model's behavior by finding important node features *e.g.,* (Xia et al., 2022), which may not suffice to capture complex temporal dependencies between nodes and temporal edges; or sampling cohesive motifs following Information Bottleneck principle (Chen & Ying, 2023), which may contain irrelevant, non-counterfactual structures for TGNN inferences. How to make the explanations accessible and responsive to users' queries is also not addressed by prior work.

**Our Contribution**. We introduce an explanatory framework to discover temporal explanations for TGNN outputs. (1) We introduce temporal graph explanations (TemGX), an instance-level structure for TGNN output (**Section 3**). A TemGX integrates a set of counterfactual, cohesive temporal subgraphs to provide "instance-level" interpretation for TGNN output. We extend counterfactual analysis (Kuratomi et al., 2024) to a sliding window to justify historical critical structures as those whose removal alter the output of TGNNs if not observed. (2) We propose a class of temporal explainability measures to quantify the quality of TemGX. Our idea extends information cascading models and resistant distance analysis, to capture the temporal influence in terms of temporal topology and node embedding similarity as an adaptive measure more suitable for evolving graphs. We show that while it is in PTIME to verify the counterfactual properties for temporal subgraphs, their generation remains NP-hard. (3) We develop efficient algorithms to discover temporal explanations for TGNN output (**Section 4**). The algorithm assembles instance-level temporal subgraphs by detecting a set of promising explanatory nodes.

Using real-world datasets, we experimentally verify the efficiency and explainability of TemGX-based TGNN explanation (**Section 5**). We show that TemGX efficiently generates that can be justified by real events, and outperform state-of-the-art TGNN explainers in terms of common measures such as fidelity. Our case study further verifies the applications of TemGX for *e.g.,* interpretable event detection in financial transaction networks, and for event detection in cybersecurity.

## 2 TEMPORAL GRAPHS AND TGNNS

**Temporal Graph**. A temporal graph $\mathcal{G}$ is a pair $(V, \mathbb{E})$, where $V$ is a finite set of nodes, and $\mathbb{E}$ is a set of temporal edges. Each temporal edge $e$ is a triple $(u, v, t)$ representing a directed interaction from a node $u$ to a node $v$ ($u, v \in V$) at time $t$ (indexed with a timestamp $t$). Each node $v \in V$ carries a label $v.L$, and a feature set $X_v$, where $X_v^t \in \mathbb{R}^d$ is the feature vector of $v$ at time $t$.

We shall use the following notations. (1) A temporal graph $\mathcal{G} = (V, \mathbb{E})$ can be represented by a discrete-time dynamic graph $\{G_1, \dots G_n\}$ (Kazemi et al., 2020), which is a sequence of snapshots. Each snapshot $G_i = (V, \mathbb{E}_i)$ has a node set $V$, and an edge set $\mathbb{E}^i \subseteq \mathbb{E}$ that contains all the temporal edges at timestamp $i$ in $\mathbb{E}$ ($i \in [1, n]$). The *size* of $\mathcal{G}$ refers to the total number of nodes and temporal edges, *i.e.,* $|V| + |\mathbb{E}|$. (2) Given a temporal graph $\mathcal{G}$ and a time window $W$ with length $t$, an temporal graph induced by $W$ is a temporal graph $\mathcal{G}_t = \{G^1, \dots, G^{t-1}\}$ of $t - 1$ consecutive snapshots (a length $t - 1$ subsequence of the snapshots) from $\mathcal{G}$, where $G^j$ refers to the $j$-th snapshot in the samples with a counterpart $G_i$ at timestamp $i$ in $\mathcal{G}$ ($i$ may not be the same as $j$). (3) Given $\mathcal{G}$, a *temporal path* $\rho_\delta$ within *duration* $\delta$ between a node $v_s$ to another node $v$ is a time-ordered sequence of temporal edges $\{(v_s, v_1, t_0), \dots (v_i, v_{i+1}, t_i), \dots (v_m, v, t_m)\}$, where $[1 \leq t_0 \leq t_m \leq t]$, and $t_m - t_0 \leq \delta$. We say $v_s$ $\delta$-*reaches* $v$ if such a path $\rho_\delta$ exists. (4) We say a temporal edge $(v, v', t')$ can be "induced" by a node set $V_\epsilon$ in $\mathcal{G}_t$, if both $v, v' \in V_\epsilon$, and $t' \leq t$. A temporal graph $\mathcal{G}_\epsilon = (V, \mathbb{E}')$ is a *temporal subgraph* of $\mathcal{G}_t = (V, \mathbb{E})$, if it has the same set of nodes $V$, and $\mathbb{E}' \subseteq \mathbb{E}$.

**Temporal Graph Neural Networks**. Following the prior study of TGNN interpretation (Kazemi et al., 2020; Xia et al., 2022; Longa et al., 2023), we consider a TGNN $\mathcal{M}$ as a composition of an encoder function $f_E$ and a decoder function $f_D$. The encoder $f_E$ takes an input temporal graph $\mathcal{G}_t = \{G^1, \dots G^{t-1}\}$ with $t - 1$ snapshots (typically induced by a time window sliding over an original temporal graph $\mathcal{G}$), and obtains $Z^{t-1}$, where $Z_v^i$ is the embedding of the node $v \in V$ at time $i$. The decoder $f_D$ takes as input $Z^{t-1}$ and output logits $Z^t$ at a next timestamp.

$$f_E(\mathcal{G}_t) = Z^{t-1}; \quad f_D(Z^{t-1}) = Z^t$$

Here the logits $Z^t(v)$ is the *output embedding* of a node $v$ at time $t$, and $\mathcal{M}(\mathcal{G}_t, v)$ (resp. $\mathcal{M}(\mathcal{G}_t, e)$) refers to an *output* of $\mathcal{M}$ at node $v$ (resp. edge $e$). The logits $Z^t$ can be post-processed to task-specific "output" such as a probabilistic matrix or discrete values. For example, $\mathcal{M}(\mathcal{G}_t, v)$ for node regression (resp. node classification) refers to a predicted values (resp. a class label), and $\mathcal{M}(\mathcal{G}_t, e)$ is a binary label for link prediction ("true" if $e$ exists in $\mathcal{G}$ at time $t$).

## 3 EXPLANATORY TEMPORAL SUBGRAPHS

A desirable explanatory structure shall be (1) responsible for TGNN output, ideally as verifiable counterfactual analysis, *i.e.,* the inference process of TGNN changes if such structures are removed from the dynamic networks; (2) capturing the temporal influence via temporal cohesive structures, and (3) support fast retrieval of temporal dependencies over time. We next introduce a class of temporal graph structures, which are justified by all the three requirements.

Given an input temporal graph $\mathcal{G}_t = (V, \mathbb{E})$ of a TGNN $\mathcal{M}$, and an output $\mathcal{M}(\mathcal{G}_t, v)$ to be explained at a target node $v$, a temporal edge $e = (u, v, t)$ is a *counterfactual edge* of $\mathcal{M}(\mathcal{G}_t, v)$, if $\mathcal{M}(\mathcal{G}_t \setminus \{e\}, v) \neq \mathcal{M}(\mathcal{G}_t, v)$; here $\mathcal{G}_t \setminus \{e\}$ is the temporal graph obtained by removing $e$ from $\mathcal{G}_t$. A node $v_s \in V$ is an *explanatory node* of $\mathcal{M}(\mathcal{G}_t, v)$, if there exists at least one of its adjacent temporal edge $e$ in $\mathcal{G}_t$ as a counterfactual edge of $\mathcal{M}(\mathcal{G}_t, v)$.

Given $\mathcal{M}(\mathcal{G}_t, v)$, and a duration constraint $\delta$, an *explanatory temporal subgraph* $\mathcal{G}_\epsilon = (V_\epsilon, \mathbb{E}_\epsilon)$ of $\mathcal{M}(\mathcal{G}_t, v)$ is a temporal subgraph of $\mathcal{G}_t$ induced by a time window of duration $\delta$, where

- $V_\epsilon = V_s \cup V_c$, where (a) $V_s$ is a set of explanatory nodes of $\mathcal{M}(\mathcal{G}_t, v)$, and (b) $V_c$ is a set of *temporal connection nodes*, such that for each $v_s \in V_s$, $v_s$ $\delta$-reaches $v$ in $\mathcal{G}_\epsilon$ via a set of connection nodes from $V_c$; here $V_s$ and $V_c$ are not necessarily disjoint, *i.e.,* an explanatory node may also be a connection node;
- $\mathbb{E}_\epsilon$ refers to the temporal edges induced by $V_\epsilon$ in $\mathcal{G}_t$.

The *counterfactual set* of $V_s$ contains all the counterfactual edges of the nodes in $V_s$.

**Justification**. We remark the following. (1) An explanatory temporal subgraph is temporally cohesive, a desirable property as for explaining GNNs in static graphs (Chen et al., 2021; Chen & Ying,

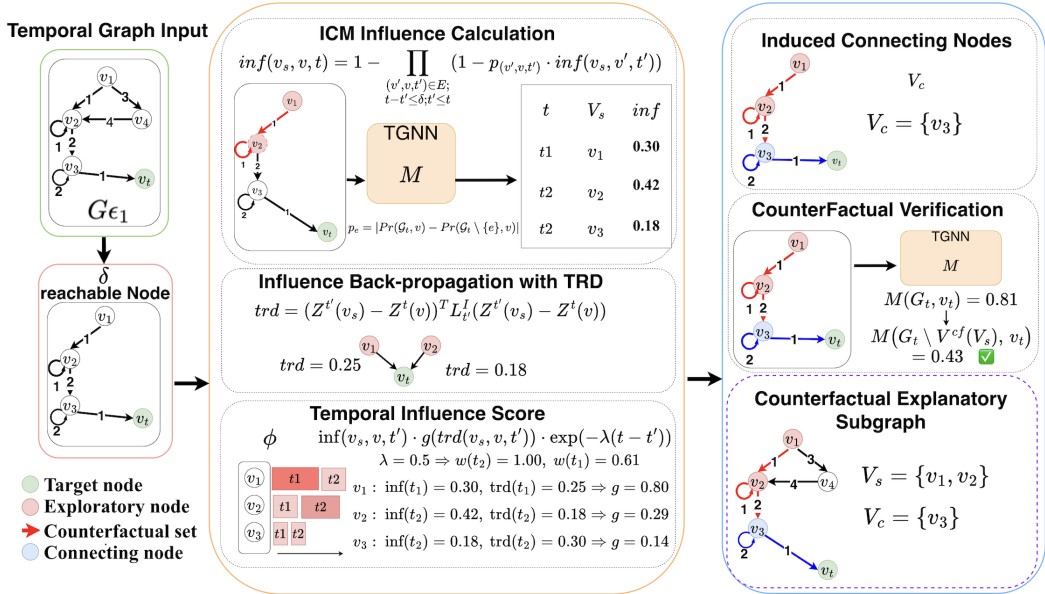

Figure 2: Overview of TemGX framework. Given a target node $v_t$ and temporal graph $G_t$, it constructs a $\delta$-reachable candidate pool and estimate influence as the probability of changes when removing each candidate. TRD back-propagation captures temporal resistance, and a temporal influence score $\phi$ combines influence, TRD, and temporal decay. Top-scoring nodes form the explanatory set $V_s$, induced connectors $V_c$ ensure temporal connectivity, and counterfactual verification confirms that removing $V_s \cup V_c$ changes the TGNN prediction $M(G_t, v_t)$, which in turn contribute to temporal explanations.

2023). This property is ensured by enforcing temporal reachability from any explanatory nodes to the target node. (2) It retains useful temporal structures that conform to counterfactual analysis to TGNN output. Unlike prior work, we distinguish "explanatory nodes" $V_s$ and "connecting nodes" $V_c$, based on a natural observation that not all nodes that (temporally) reach $v$ are necessarily "influencing" the decision of TGNNs. Indeed, those that may be responsible for TGNN output can be temporarily remote ("from the past") and structurally non-adjacent with $v$. (3) They readily support both temporal graph queries, to extract "instance-level" explanations, and probabilistic inference via *e.g.,* dynamic Bayesian networks, to understand temporal dependencies over time (see Section 4).

**Temporal Explainability**. We next quantify the explainability of an explanatory temporal subgraph $G_\epsilon$ in terms of its temporal influence to a targeted output $\mathcal{M}(\mathcal{G}_t, v)$, considering both temporal propagation and the node embedding similarity, consistent with TGNN inference process.

(1) We formulate temporal impact to TGNN output, by extending the Independent Cascade Model (ICM), which models influence propagation through independent edge activations with probability $p(v, v', t)$ at time $t$. Given $G_\epsilon$ and a target output $\mathcal{M}(\mathcal{G}_t, v)$, the temporal influence of a temporal edge $e$ at time $t$ on node $v$ is defined as:

$$p_e = \left| Pr(\mathcal{G}_t, v) - Pr(\mathcal{G}_t \setminus \{e\}, v) \right|$$

where $Pr(\mathcal{G}_t, v)$ is the task-specific output probability from the embedding $Z^t(v)$ and $Pr(\mathcal{G}_t \setminus \{e\}, v)$ is the probability *if* $e$ is removed at time $t$ [1]. The influence of an explanatory node $v_s$ on $v$ recursively aggregates along all $\delta$–reachable temporal paths:

$$\text{inf}(v_s, v, t) = 1 - \prod_{\substack{(v', v, t') \in E \\ t - \delta \leq t' \leq t}} \left( 1 - p_{(v', v, t')} \cdot \text{inf}(v_s, v', t') \right)$$

We set $\text{inf}(v_s, v_s, t) = 1$ to ensure that an explanatory node has full influence to itself.

(2) An TGNN inference process propagates node embeddings via temporal paths to output for decision making. This effect can be approximated by a class of *temporal effective resistance distance*, which has a justified static counterpart in GNNs behavior analysis (Zhu et al., 2024; Black et al., 2023) and model optimization (Shen et al., 2024). Given a pair $(v_s, v)$ of an explanatory node $v_s$ and a targeted node $v$, we define the temporal resistance distance at time $t'$ as:

$$\text{trd}(v_s, v, t') = (Z^{t'}(v_s) - Z^t(v))^T L^I_{t'} (Z^{t'}(v_s) - Z^t(v))$$

---

[1] The absolute value captures magnitude since edge removal can increase or decrease the probability.

where $Z^{t'}(v_s)$ (resp. $Z^t(v)$) refers to the embedding of $v_s$ at time $t' \leq t$ (resp. the output embedding of $v$ at time $t$). Let $L_{t'}$ be the Laplacian matrix of the snapshot $G_{t'} \in \mathcal{G}_t$ (i.e., $L_{t'} = D_{t'} - A_{t'}$; where $D_{t'}$ and $A_{t'}$ refers to the degree matrix and the adjacency matrix of snapshot $G_{t'}$ with eigenvalues $\lambda_1, \ldots \lambda_n$ and corresponding eigenvectors $u_1, \ldots, u_n$, the Laplacian pseudo inverse $L_{t'}^I$ at time $t'$ is computed as $\sum_{l=2}^n \frac{1}{\lambda_l} u_l \, u_l^T$. We adopt efficient algorithms such as (Liao et al., 2023) to approximate $\mathsf{trd}(v_s, v, t)$ in low polynomial time.

The *temporal influence* of a set of explanatory nodes $V_s$ over $[t - \delta + 1, t - 1]$ is aggregated as

$$\Phi(V_s, v) = \frac{1}{\delta} \sum_{v_s \in V_s} \sum_{t' = t - \delta + 1}^{t-1} \inf(v_s, v, t') \, g(\mathsf{trd}(v_s, v, t')) \, e^{-\lambda(t - t')}$$

where $g : \mathbb{R}_{\geq 0} \to [0, 1]$ is a monotone decreasing function of the temporal resistance distance and $\lambda > 0$ controls exponential time decay. The multiplicative form lets temporal resistance modulate rather than cancel influence, while $e^{-\lambda(t-t')}$ discounts older interactions so recent ones weigh more (Mei & Eisner, 2017). Because $\inf \in [0, 1]$, $g \in [0, 1]$, and the decay term is in $(0, 1]$ for $t' < t$, $\Phi(V_s, v) \in [0, 1]$. A larger $\Phi$ indicates stronger, more recent influence under $\delta$–reachability. The decay parameter $\lambda$ is scale–adaptive, typically inversely proportional to the temporal granularity (Rossi et al., 2020a). Note that cohesiveness does not require consecutive temporal edges.

**Explanation Generation for TGNNs**. Based on the explainability measures, we formulate TGNN explanation problem. Given a set of targeted nodes $V_T$, with target outputs $\mathcal{M}(\mathcal{G}_t, v)$ $(v \in V_T)$, and a duration bound $\delta$, for each node $v \in V_T$, the problem is to find an optimal set of explanatory temporal subgraphs $\eta$ induced by a set of explanatory nodes $V_s^*$, such that

$$V_s^* = \arg \max_{|V_s| \leq k} \sum_{v \in V_T} \sum_{v_s \in V_s} \Phi(V_s, v);$$

where $v$ ranges over $V_T$, $V_s^* = \cup_{v \in V_T} V_s^*$, and $V_s^*$ is the set of all the explanatory nodes for a target output $\mathcal{M}(\mathcal{G}_t, v)$. In addition, a size bound $k$ constraints the total number of explanatory nodes, to allow users specify concise, size-bounded instance-level explanations.

We investigate the hardness of the problem. We start with a *verification* problem as a "building block" task. Given an output $\mathcal{M}(\mathcal{G}_t, v)$ to be explained, a duration bound $\delta$, and a set of nodes $V_s$, it is to verify if $V_s$ induces an explanatory temporal subgraph with temporal explainability above a given threshold. Successful verification of $V_s$ indicates the existence of instance-level explanations and a proper interpretable domain $\eta$.

**Lemma 1:** *The verification problem is in* PTIME.

We prove the above result by constructing a PTIME procedure (Verify). It computes influence scores of a given temporal subgraph $G_s$ to be verified, and invokes TGNN inference process to verify if its counterfactual set is empty. Despite the tractability of verification, the overall explanation discovery problem is nontrivial.

**Lemma 2:** TGNN *explanation generation problem is* NP-*hard.*

The hardness of the low-level optimization can be verified by a reduction from the maximum coverage problem (MCP), a known NP-hard problem.We present the detailed proof in Appendix 7. Despite the hardness, we next introduce efficient algorithms to generate temporal explanations with guaranteed temporal explainability.

## 4 DISCOVERING TEMPORAL GRAPH EXPLANATIONS

We start with a setting for a single output $\mathcal{M}(\mathcal{G}_t, v)$. Our algorithm, denoted as TemGX, follows a "selection-and-verify" process to grow a set of exploratory $\mathcal{V}_s$ that simultaneous survive a counterfactual checking posed on their temporally adjacent edges.

**Algorithm Outline**. TemGX applies a greedy selection procedure to iteratively expand the explanatory set $V_s$ inside a sliding window $W$. It begins with $V_s = \{v\}$ and a candidate pool $C$ formed by the $L$–hop temporal neighbors $N_L(v)$ within the induced temporal graph $G_W$. At each iteration, the algorithm selects the node $v_s^*$ in $C$ that maximizes the temporal influence score $\Phi(v_s, v)$, updates the influence scores of remaining candidates, and invokes the Verify procedure to ensure that $v_s^*$ preserves the counterfactual property of the TGNN output. The verified nodes are appended to $V_s$,

---

**Algorithm** TemGX $(\mathcal{G}_t, \mathcal{M}, \mathcal{M}(\mathcal{G}_t, v), k)$

1. set $\eta := \emptyset$; set $\mathcal{C} := \emptyset$;
2. **for each** window $W$ of size $\delta$ over $\mathcal{G}_t$ **do**
3.     set $V_s := \{v\}$; set $\mathcal{C} := \emptyset$; induce a temporal graph $G_W$ with current window $W$;
4.     **for each** time step $t_i$ in $W$ **do** induce $L$-hop temporal neighbor set $N_L(v)$ in $G_W$;
5.         **for each** $v' \in N_L(v)$ **do** update $v'.\Phi := \Phi(v', v)$; $\mathcal{C} := \mathcal{C} \cup \{v_s\}$;
    */* select and verify explanatory nodes */*
6.     **while** $\mathcal{C} \neq \emptyset$ **do** */* select and verify explanatory nodes */*
7.         select node $v_s^* = \arg\max_{v_s \in \mathcal{C}} \Phi(v_s, v)$; updatePhi($\mathcal{C}$);
8.         **if** Verify$(\mathcal{M}, \mathcal{M}(\mathcal{G}_t, v), V_s, v)$ = false **then** $\mathcal{C} := \mathcal{C} \setminus \{v_s^*\}$; **continue**;
9.         **if** $|V_s| < k$ **then** $V_s := V_s \cup \{v\}$; $\mathcal{C}^t := \mathcal{C}^t \setminus \{v_s\}$; **break**;
10.         **else** set Rset $:= V_s$; */* replacement policy */*
11.           **while** Rset $\neq \emptyset$ **do** select node $v_s^{*'}$ as $\arg\max_{v_s' \in \text{Rset}} \Phi(V_s \setminus \{v_s^{*'}\} \cup \{v_s^*\})$;
12.           **if** Verify$(\mathcal{M}, \mathcal{M}(\mathcal{G}_t, v), V_s \setminus \{v_s^{*'}\} \cup \{v_s^*\}, v)$ = false
13.             **then** Rset:=Rset $\setminus \{v_s^{*'}\}$; **continue**;
14.           **else** $V_s := V_s \setminus \{v_s^{*'}\} \cup \{v_s^*\}$; $\mathcal{C} := \mathcal{C} \setminus \{v_s\}$; **break**;
    */* construct a temporal explanatory subgraph */*
15.     induces $\mathcal{G}_\epsilon$ with $V_s$, $V_c$ (connect nodes) and $\mathbb{E}_\epsilon$; $\eta := \eta \cup \{\mathcal{G}_\epsilon\}$;
16. **return** $\eta$;

Figure 3: Algorithm TemGX

whereas failed candidates are removed from $C$. When $|V_s|$ reaches the budget $k$, a replacement policy is triggered: it searches a temporary set Rset of nodes in $V_s$ and greedily swaps a node $v_s^{*'}$ in Rset with a candidate $v_s^*$ if the exchange yields a larger marginal influence gain $\Phi(V_s \setminus \{v_s^{*'}\} \cup \{v_s^*\}, v)$ and passes verification. This process continues until the candidate pool is empty, and returns the temporal explanatory subgraphs $G_\epsilon$ with connecting nodes $V_c$ and temporal edges $E_\epsilon$.

**Analysis**. Denote the optimal set of explanatory nodes as $V_s^O$, we present the following result.

**Lemma 3:** *Algorithm* TemGX *is a* $(1 - \frac{1}{e})$-*approximation,* i.e., $\Phi(V_s, v) \geq (1 - \frac{1}{e})\Phi(V_s^O, v)$.

We prove the above result by (1) reducing the problem to an equivalent online monotone submodular maximization problem with a cardinality constraint over a stream of temporal edges, and (2) proving that the greedy replacement policy ensures a $(1-\frac{1}{e})$-approximation for $\Phi$ as a monotone submodular function (Qian et al., 2017). For **time cost**, it takes TemGX at most $t$ rounds of node selection, each in $O(|N_L(v)|T_I)$ time to compute the temporal influence score, where $T_I$ is the polynomial-time inference cost of the TGNN $\mathcal{M}$. We remark that TemGX is a "one-pass" algorithm: the temporal nodes and their temporal neighbors are visited and processed only once as the sliding window moves forward. We present the detailed analysis in Appendix 7.

**Querying Temporal Explanations**. TemGX provides a natural query access point with semantics of temporal pattern matching (Ding et al., 2023; Chen & Ying, 2023). (1) A temporal pattern $Q_\eta = (V_Q, E_Q, \sigma, \delta')$ specifies a set of query nodes $V_Q$ and query edges $E_Q$. Each query node $v_q \in V_Q$ may carry a label $v_q.L$. Here $\sigma$ is a temporal order posed on $E_Q$. Given a set of temporal explanatory subgraphs $\eta$ with a node set $V_\eta$, and a query $Q_\eta = (V_Q, E_Q, \sigma, \delta')$, there exists a *match* of $Q$ in $\eta$, if there is a mapping $h : V_Q \to V_\eta$, such that for each edge $e_q^i = (u_i, u_{i+1})$ in $Q$, (i) $h(u_i) = v_i$, $u_i.L = v_i.L$; (ii) $h(u_{i+1}) = v_{i+1}$, $u_{i+1}.L = v_{i+1}.L$; and (iii) $v_i$ $\delta'$-reaches $v_i'$. Such queries can be efficiently processed by temporal pattern matching algorithms *e.g.,* (Li & Zou, 2021; Aghasadeghi et al., 2024). (2) Better still, TemGX supports a *statistical inference* analysis via an efficient summary that learns a dynamic Bayesian network. We defer the details in Appendix 7.

## 5 EXPERIMENTAL STUDY

We evaluate the explainability and efficiency of TemGX. Our code and datasets are available[2].

### 5.1 EXPERIMENTAL SETUP

**Models and Datasets**. We evaluate on six real-world temporal graphs. (1) For spatio-temporal regression, we use METR-LA and PEMS-BAY (Li et al., 2018). For link prediction, we use Wiki (Ku-

---

[2]https://github.com/nicej1899/TemGX/

Table 1: AUFSC, Fidelity, and Runtime (seconds) of generating one explanation on each dataset/backbone (mean ± std).

| Link Prediction | | TemGX | | | TempME | | | TGNNExplainer | | | CoDy | | |
|---|---|---|---|---|---|---|---|---|---|---|---|---|---|
| Dataset | Model | AUFSC | Fid | Time | AUFSC | Fid | Time | AUFSC | Fid | Time | AUFSC | Fid | Time |
| UCIM | TGN | **0.475** ±0.012 | **0.468** ±0.014 | **8.2** ±0.19 | 0.218 ±0.013 | 0.219 ±0.015 | 88.4 ±1.7 | 0.073 ±0.014 | 0.072 ±0.016 | 68.5 ±2.0 | 0.361 ±0.018 | 0.394 ±0.021 | 35.0 ±1.8 |
| | TGAT | **0.423** ±0.011 | **0.415** ±0.010 | **7.8** ±0.17 | 0.153 ±0.012 | 0.156 ±0.013 | 84.1 ±1.6 | 0.191 ±0.013 | 0.187 ±0.015 | 65.2 ±1.9 | 0.349 ±0.020 | 0.382 ±0.024 | 38.4 ±2.0 |
| Wikipedia | TGN | **0.266** ±0.016 | **0.262** ±0.016 | **12.5** ±0.34 | 0.128 ±0.015 | 0.130 ±0.016 | 72.8 ±1.9 | 0.098 ±0.017 | 0.096 ±0.018 | 65.2 ±2.1 | 0.200 ±0.021 | 0.219 ±0.024 | 125.0 ±4.8 |
| | TGAT | **0.340** ±0.018 | **0.335** ±0.019 | **11.8** ±0.33 | 0.155 ±0.016 | 0.154 ±0.017 | 75.4 ±2.0 | 0.117 ±0.018 | 0.115 ±0.018 | 68.6 ±2.3 | 0.184 ±0.022 | 0.201 ±0.025 | 108.2 ±4.3 |

| Spatio–Temporal Regression | | TemGX | | | TGVex | | |
|---|---|---|---|---|---|---|---|
| Dataset | Model | AUFSC | Fid | Time | AUFSC | Fid | Time |
| METR-LA | STGCN | **0.478** ±0.009 | **0.471** ±0.011 | **6.1** ±0.12 | 0.276 ±0.010 | 0.276 ±0.011 | 6.8 ±0.15 |
| | DCRNN | **0.443** ±0.010 | **0.436** ±0.012 | **6.4** ±0.11 | 0.258 ±0.011 | 0.258 ±0.011 | 6.6 ±0.14 |
| PEMS-BAY | STGCN | **0.480** ±0.008 | **0.473** ±0.008 | **6.3** ±0.10 | 0.277 ±0.009 | 0.277 ±0.010 | 6.7 ±0.13 |
| | DCRNN | **0.448** ±0.009 | **0.441** ±0.009 | **6.7** ±0.10 | 0.261 ±0.010 | 0.261 ±0.010 | 6.8 ±0.13 |

mar et al., 2019) and UCIM (Kunegis, 2013). For classification, we use Multihost (Gao et al., 2021) and Elliptic++ (Elmougy & Liu, 2023) . Dataset details are provided in Appendix 7. (2) We have trained four types of TGNNs. (1) STGCN (Yu et al., 2018) employs graph convolution to capture spatial dependencies and temporal convolution to model temporal patterns. (2) DCRNN (Li et al., 2018) captures spatial dependencies through graph diffusion and temporal dynamics via recurrent units, enabling accurate spatio-temporal forecasting. (3) TGN (Rossi et al., 2020b) uses memory modules and attention mechanisms to capture the temporal evolution of node states. (4) TGAT (Xu et al., 2020) combines self-attention to learn spatial dependencies. We trained STGCN, DCRNN for METR-LA, PEMS-BAY; and TGN and TGAT for Wiki and UCIM, uniformly with 70%, 15% and 15% for training, validating, and testing.

**TGNN Explainers**. We have implemented the following. (1) TemGX explainer; (2) TGVex, an extension of GVex (Chen et al., 2024) by applying it over "static" snapshots induced by a sliding window to generate a sequence of explanatory subgraphs; (3) TGNNExplainer (Xia et al., 2022), an explorer-navigator framework to explain TGNN-based link (event) inference; (4) TempME (Chen & Ying, 2023), a temporal motif-based explainer that finds events in terms of information bottleneck scores. (5) CoDy (Qu et al.), a counterfactual explainer that leverages Monte Carlo Tree Search to efficiently identify minimal event subsets whose removal flips the TGNN prediction. Among these, TempME and TGNNExplainer are learning-based explainers; CoDy uses both search and training and all are model-specific hence their time cost include ad-hoc training overhead. TGNNExplainer, CoDy and TempME are designed for event explanation and generate edges only. Hence, we compared TemGX with them using Wiki and UCIM for event analysis.

## 5.2 EXPERIMENTAL RESULTS

**Exp-1: Explainability**. We report the fidelity and sparsity of all explainers.

*Fidelity: Overall.* Table 1 reports the fidelity of all methods under the same evaluation protocol. For Wiki and UCIM, we set $k = 50$ and $|V_T| = 500$ and we use both TGN and TGAT backbones for link prediction tasks. For METR-LA and PEMS-BAY, we adopt STGCN and DCRNN for spatio–temporal regression tasks. (1) TemGX achieves the best fidelity across all datasets and models. Its advantage stems from the use of temporal influence scores and the counterfactual property, which capture dynamic dependencies missed by snapshot-based approaches. For example, on UCIM with TGN, TemGX attains a fidelity of **0.468**, surpassing TempME (0.219) by 113%, 6.5× higher than TGNNExplainer (0.072), and still 19% higher than the strong CoDy baseline (0.394). On the TGAT backbone, TemGX improves fidelity to **0.415**, exceeding TempME (0.156) by 166%, 2.2× higher than TGNNExplainer (0.187), and outperforming CoDy (0.382) by 9%. On Wiki, TemGX obtains the highest fidelity on both backbones (e.g., **0.262** on TGN and **0.335** on TGAT), surpassing CoDy (0.219 / 0.201) by about 20% and 67%, and remaining far ahead of TempME (0.128 / 0.155) and TGNNExplainer (0.098 / 0.117). For regression tasks, TemGX also leads: it improves over TGVex by about 20% on METR-LA (DCRNN) and 18% on PEMS-BAY (DCRNN), with similar margins on the TGCN backbone. (2) TGVex performs poorly because its snapshot–pattern mining ignores $\delta$–reachability and dynamic temporal dependencies that drive TGNN inference. (3) TGNNExplainer struggles in particular on dense graphs such as Wiki, where its edge–centric Monte Carlo search cannot efficiently identify nodes with strong accumulated temporal influence. (4) TempME benefits from motif discovery and achieves competitive scores on some link–prediction settings, but mainly captures frequent motifs and often misses less frequent yet highly influential temporal patterns.

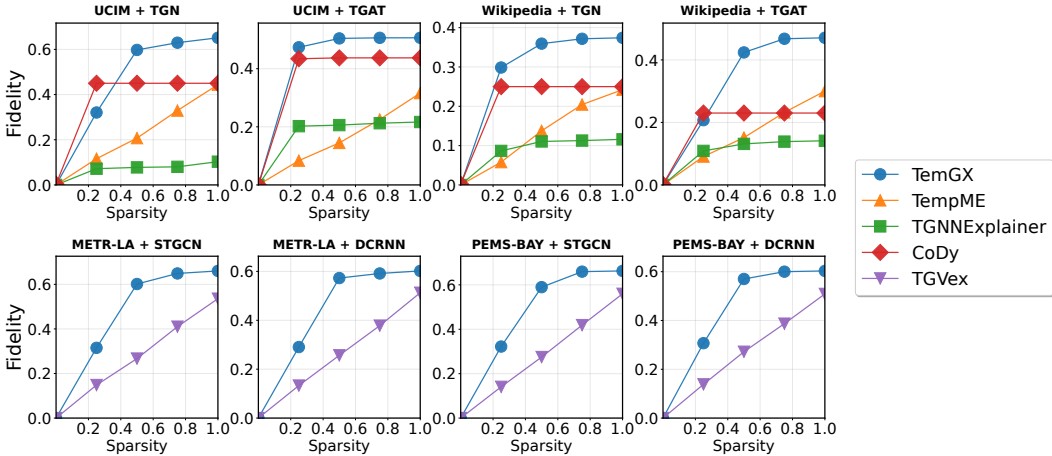

Figure 4: Fidelity-Sparsity Trade off.

AUFSC: *Overall*. Table 1 reports the AUFSC scores under the same setting. TemGX again achieves the best results across all datasets and models. On UCIM (TGN) it reaches an AUFSC of **0.475**, more than $2.2\times$ that of TempME (0.218), over $6.5\times$ higher than TGNNExplainer (0.073), and still 31% higher than the strong counterfactual baseline CoDy (0.361). On the TGAT backbone, TemGX achieved AUFSC to **0.423**, exceeding TempME (0.153) by 176%, $2.2\times$ higher than TGNNExplainer (0.191), and outperforming CoDy (0.349) by a clear 21% margin. On Wiki, TemGX maintains clear superiority, achieving AUFSC of **0.266** (TGN) and **0.340** (TGAT), surpassing CoDy (0.200 / 0.184) by about 33% and 85%, and remaining far ahead of both TempME (0.128 / 0.155) and TGNNExplainer (0.098 / 0.117). For spatio–temporal regression tasks, TemGX also consistently dominates TGVex, with the largest margin (about 65%) observed on PEMS-BAY (DCRNN). *Fidelity–sparsity trade-off curves*. Using Wiki with $|V_T| = 500$ as in Fig. 4, we report the Fidelity–Sparsity trade-off curves of TemGX in Fig. 4. These curves illustrate how the fidelity of TemGX evolves as its explanation size varies, providing insight into the internal sparsity–fidelity behavior of our method.

**Exp-2: Efficiency**. We also report in Table 1 the time required to generate a single explanation. (1) TemGX is consistently fast: on link–prediction datasets it completes in 12.5 s, and on regression datasets, within 6.7 s. (2) On the dense Wiki graph, TemGX delivers $5.2\times$–$6.4\times$ speedups over TGNNExplainer (65.2/68.6 s) and TempME (72.8/75.4 s), and up to $10\times$ over CoDy (125.0/108.2 s), while maintaining the highest AUFSC and fidelity. On UCIM, TemGX is $8.3\times$–$8.4\times$ faster than TGNNExplainer, $10.8\times$ faster than TempME, and about $4.3\times$–$4.9\times$ faster than CoDy. (3) TGVex is competitive with TemGX on the sparser METR-LA and PEMS-BAY networks, but at the cost of lower fidelity (see Fig. 4). (4) Training cost matters: TGNNExplainer (navigator) and TempME (motif scorer) are learning–based and require an extra training phase before explaining, whereas both TemGX and CoDy are *training–free* search methods operating directly on a pre–trained TGNN. Given similar training-free settings, TemGX attains substantially better AUFSC/Fidelity than CoDy while also being much faster, which further widens the efficiency gap in practice.

**Exp-3: Case Analysis**. We next demonstrate how TemGX enables queryable explanations for real-world TGNN tasks. (1) **Clarifying Suspicious Transactions.** Fig. 5(i) shows a Bitcoin subgraph centered on node 140 (an "illicit" IP detected by a TGN). Fig. 5(ii) presents three temporal explanatory subgraphs $\mathcal{G}_{\epsilon_1}$–$\mathcal{G}_{\epsilon_3}$ produced by TemGX for node 140, aligned with known patterns (blue dashed circles for "Spindle", red for "Peel Chain"). IPs 418, 2819, 2411, and 140 form a Spindle match at the first timestamp; next, 418 is replaced by 782 with 2558 as new Spindle participants; later, a "Peel Chain" involving 2216 emerges. These subgraphs can be directly queried to retrieve suspicious matches of either pattern for follow-up investigation. (2) **Multi-stage Cyber Attack Analysis.** Fig. 5(b)(i) showcase a three-stage attack from Multihost: compromises via Shellshock (host1 executes gather_password.sh, identifies and sends crack_passwd.sh) for password cracking on host2 (downloading malicious "libfoo.so"), and data exfiltration (using an MD5 hash to extract "john.zip"). Two temporal explanatory subgraphs found by TemGX (Fig. 5(b)(ii)) accurately

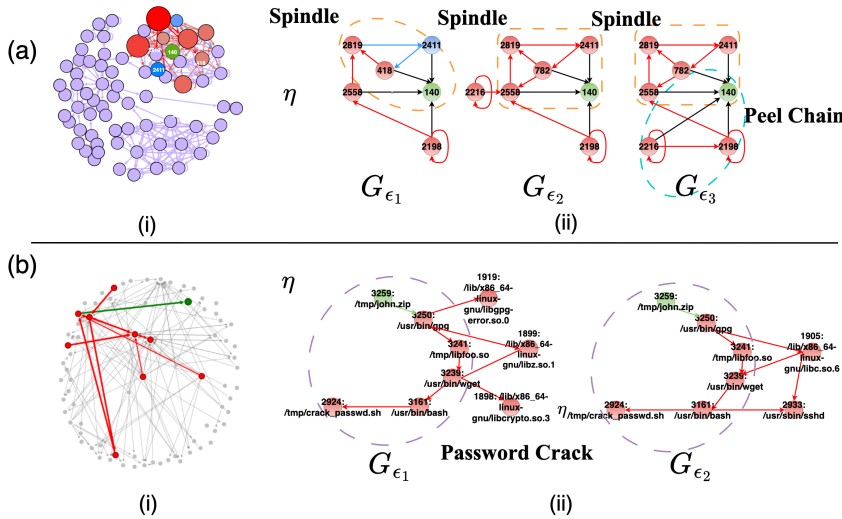

Figure 5: Case Analysis

cover the critical host1–host2 paths, including nodes and connections with gather_password.sh and crack_passwd.sh, illustrating TemGX 's ability to explain complex multi-step attacks.

# 6 RELATED WORK

*Temporal Graph Neural Networks.* TGNNs extends GNNs by integrating graph convolutions with temporal dependency of dynamic patterns. TGNNs can effectively capture both spatial dependencies and temporal sequences in a dynamic network, leading to more accurate predictions of future graph states (Longa et al., 2023). The temporal inference process operates on a sequence of temporal features to produce a prediction of the expected output at the subsequent times. Notable TGNNs include TGCNs (Yu et al., 2018), DCRNNs (Li et al., 2018), among others (Xu et al., 2020).

*Graph Neural Network Explanation.* Among the relevant approaches are *post-hoc* methods (Funke et al., 2023; Luo et al., 2020; Vu & Thai, 2020; Ding et al., 2023; Chen et al., 2024; Prenkaj et al., 2024), which create separate models to generate explanations for a GNN. Notably, PGExplainer (Luo et al., 2020) extracts important features and substructures that influence GNN predictions. (Vu & Thai, 2020) generates a Bayesian network that best fits a set of subgraphs that influences GNN output. (Ding et al., 2023) generate motifs as explanations. (Chen et al., 2024) generates explanatory views that contain graph patterns as explanatory structures. For TGNNs, (He et al., 2022) extends (Vu & Thai, 2020) to learn an optimal Bayesian networks over sliding windows. TGNNExplainer (Xia et al., 2022) pretrains a navigator (a feed-forward neural network) that infers importance scores among events, and an explorer that uses Monte Carlo Tree Search to infer events that minimize a cross entropy loss to target events. TempME (Chen & Ying, 2023) samples motifs and learns to infer their importance in terms of the information bottleneck scores as explainability measure. CoDy (Qu et al.) employs a Monte Carlo Tree Search–based strategy to search for minimal event subsets whose removal alters TGNN output. (Vu & Thai, 2022) analyzed the limitations of perturbation-based TGNN explainers and proposed temporal perturbation as a direction for faithful temporal reasoning. Our work is among the first to unify counterfactual analysis, instance-level temporal explainability and continuous query into an efficient framework for TGNNs.

# 7 CONCLUSIONS

We have presented TemGX, a novel framework for TGNN output explanation. We introduced temporal graph explanation (TemGX), that integrates instance-level temporal explanation subgraphs to capture their temporal dependencies over time. We have developed fast algorithms to generate TemGX. Our experimental study has verified that TemGX outperforms state-of-the-art TGNN explainers in explainability and efficiency. A future topic is to optimize TemGX to scale TGNN interpretation over large distributed networks and real-time graph stream analysis.

ETHICS STATEMENT

All authors of this submission adhere to the ICLR Code of Ethics. Our work does not involve human subjects, sensitive personal data, or high–risk applications. Potential concerns such as privacy, security, or legal compliance have been carefully considered and are not applicable. To the best of our knowledge, no conflicts of interest, discrimination, or harmful insights arise from the methods or datasets used in this paper.

REPRODUCIBILITY STATEMENT

We have made every effort to ensure the reproducibility of our results. All models, hyperparameters, and training procedures are described in the main text and Appendix. An anonymized repository containing source code and datasets is provided in the supplementary materials and via a github link `https://github.com/nicej1899/TemGX/`, allowing independent verification of all tests. Detailed proofs of the theoretical claims are included in the Appendix, and all preprocessing scripts are supplied to facilitate replication of the reported results.

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

APPENDIX

## A  DETAILS OF THE ALGORITHMS

**Algorithm** TemGX (Fig 3) takes as input a temporal graph $G^t$, a TGNN model $M$, a target output $M(G^t, v)$ to be explained, and a budget parameter $k$. It works with a sliding window of length $\delta$ over $G^t$ to generate a set of instance-level explanations $\eta$. For each window $W$, it generates a temporal explanatory subgraph with at most $k$ explanatory nodes through a selection and verify process.

**(1) Initialization and Window Setup (lines 1-5).** TemGX initializes the result set $\eta = \emptyset$ and candidate pool $C = \emptyset$ (line 1), then processes each sliding window $W$ of size $\delta$ over the temporal graph $G^t$ (line 2). For each window, it sets the explanatory set $V_s = \{v\}$ starting with the target node, resets the candidate pool $C = \emptyset$, and induces a temporal subgraph $G_W$ with the current window $W$ (line 3). For each time step $t_i$ in $W$, it induces the $L$-hop temporal neighbor set $N_L(v)$ in $G_W$ (line 4). For each node $v' \in N_L(v)$, it computes the temporal influence score $v'.\Phi := \Phi(v', v)$ and adds to the candidate pool $C := C \cup \{v_s\}$ (line 5). This exploits spatiotemporal data locality to constrain the search space to nodes that may participate in TGNN $M$ inference under duration constraint $\delta$.

**(2) Greedy Selection and Replacement (lines 6-14).** TemGX adopts a greedy selection and replacement policy through "one-pass" processing of $C$. The main selection loop continues while the candidate pool is non-empty (line 6). At each iteration, it selects the candidate node $v_s^*$ that maximizes the temporal influence score $\Phi(v_s, v)$ over all $v_s \in C$ and updates influence scores of remaining candidates via updatePhi($C$) (line 7). It invokes the Verify procedure with model $M$, target output $M(G^t, v)$, current explanatory set $V_s$, and target node $v$ (line 8)—if verification fails, $v_s^*$ is removed from $C$ and the loop continues. When $|V_s| < k$, the algorithm adds $v_s$ to $V_s$ and removes $v_s$ from $C^t$, then breaks (line 9). For $|V_s| = k$, it switches to a replacement strategy (line 10): it sets $Rset = V_s$ and selects node $v_s^{*'}$ that maximizes $\Phi(V_s \setminus \{v_s^{*'}\} \cup \{v_s^*\})$ (line 11). If verification of the replacement with model $M$ fails, $v_s^{*'}$ is removed from $Rset$ (lines 12-13); otherwise, the replacement is executed: $V_s := V_s \setminus \{v_s^{*'}\} \cup \{v_s^*\}$ and $v_s$ is removed from $C$ (line 14). This process repeats until $C$ is completely processed.

**(3) Explanation Construction (lines 15-16).** Once the selection process completes, the algorithm constructs temporal explanatory subgraph $G_\epsilon$ by inducing it with explanatory nodes $V_s$, connecting nodes $V_c$, and edge set $E_\epsilon$ (line 15). The constructed explanation $G_\epsilon$ is added to the result set $\eta := \eta \cup \{G_\epsilon\}$. The algorithm processes the next window and terminates when all sliding windows are consumed, returning the complete set $\eta$ of discovered temporal explanatory subgraphs (line 16).

## B  TEMPORAL EXPLANATORY SUBGRAPHS: SUMMARIES AND QUERYING

We elaborate how TemGX support cost-effective and fast query access and statistical inference of temporal explanatory subgraphs. TemGX employs Dynamic Bayesian Networks (DBN) (Murphy, 2002) as a summary structure to encode the dynamic trajectory of the explanatory nodes and their temporal edges. The summarization process serves as a necessary statistical modeling step that allows users to perform temporal reasoning and "What-if" analysis over the discovered explanations.

**Summarization Structure.** Temporal dependencies arise from the interactions among explanatory nodes and sequences of explanatory temporal subgraphs. Probabilistic graphical models such as dynamic Bayesian networks (DBN) (Murphy, 2002) provide a concise and expressive model to capture temporal dependencies. To summarize and enable querying of instance-level explanations over time, we introduce a DBN-based summarization structure.

**Structure Definition.** Given a set of temporal explanatory subgraphs $\eta = \{\mathcal{G}_\epsilon^1, \ldots \mathcal{G}_\epsilon^n\}$ over time $[1, t-1]$ as discovered explanations for $\mathcal{M}(\mathcal{G}_t, v)$, a *temporal explanation summary* $\mathcal{P}$ is a directed acyclic graph (DAG) which contains a set of variables $\mathcal{X}^i$ for each timestamp $i$ ($i \in [1, t]$), where:

(1) Each variable $x_s^i$ in $\mathcal{P}$ corresponds to an explanatory node $v_s^i$ at time $i$ from a subgraph in $\eta$, and is associated with a discrete, temporal *state* from a finite state set $S$ via a state encoding function from $v_s^i$ to $x_s^i$. The state set at timestamp $i$ is denoted as $X_i$, which may encode node label in node

classification, or edge labels for link prediction. Each variable $x_s^i$ carries a conditional probability table $x_s^i.T$ to bookkeep its temporal dependencies with its neighboring variables.

(2) There are two types of edges: (a) An "intra-slice" edge outlines the conditional dependencies between two variables from a subgraph in $\eta$ at time $i$, and (b) a set of "inter-slice" edges, connecting variables across time stamps, with their directions aligned with the progression of time. These edges help track temporal state changes as the variables evolve, capturing the dynamic nature of the temporal subgraphs.

**Summary Learning Objective**. Constructing summary $\mathcal{P}$ over a given set of temporal explanatory subgraphs can be consistently formulated as learning an optimal dependency structure of a DBN (Koller & Friedman, 2009). The learning objective of $\mathcal{P}$ can be formulated as optimizing a unified Bayesian Information Criterion (BIC) that optimizes both intra-slice and inter-slice dependencies. Given $\eta$ and a DBN $\mathcal{P}$, the BIC score is computed as follows:

$$\psi(\mathcal{P}, \eta.S) = \mathsf{LL}(\eta.S|\mathcal{P}) - \frac{|\mathcal{P}|}{2} \log |\eta.S|$$

where $\mathsf{LL}(\eta.S|\mathcal{P})$ is the log-likelihood of the states $\eta.S$ given $\eta$ and the learned structure of $\mathcal{P}$, $|\mathcal{P}|$ is the number of the free parameters in the learned structure of $\mathcal{P}$, and $|\eta.S|$ is the number of observed states associated to the nodes in $\eta$. Given $\eta$, TemGX learns a summary $\mathcal{P}$ with a state encoding and structure such that $\psi$ is optimized.

**Summary Generation Process**. For each explanatory node $v_s \in V_s$ at timestamp $i$, the summarization process extracts feature embeddings from temporal snapshots and discretizes the values using uniform binning to ensure a finite state representation. This yields a structured discrete state set $\eta.S$ to encode the temporal distribution of $V_s$ in a probabilistic space.

The process then learns a set of intra-slice edges and inter-slice edges for $V_s$ and $v$, using BIC score evaluated on aggregated data from all consecutive timestamp pairs $[i, i+1]$ for $i \in [1, t–2]$. To this end, it invokes fast heuristics (Gámez et al., 2011) that ensures a competitive local optima. It finally performs a Maximum Likelihood Estimation to optimize the conditional probability distributions of the state nodes, to ensure $\mathcal{P}$ accurately fits the observed $\eta$.

**Query Support via DBN Summarization**. The learned DBN summary $\mathcal{P}$ naturally supports inference analysis for temporal reasoning. One type of useful inference analysis is to request the likelihood of a value of a variable $x_u$ of interests in $\eta$ at a particular time $t' \in [1, t]$ (where the value may not be seen in $\mathcal{G}_t$), given a confirmed occurrence of the value of another node variable $x_u'$ at another time $t'' \in [1, t]$. This allows efficient "What if" questions.

For example, a "What-if" question ask "*How likely $v_t$'s label becomes 'licit' (indicating that TGNN makes a mistake) **if** the label of $v_1$ is enforced to be 'licit', given discovered explanations $\eta$?*" The inference analysis can be performed by efficient DBN inference process (Murphy, 2002), with time cost in $O(|\eta|t)$ time, where $|\eta|$ is the total number of nodes and edges of the temporal subgraphs in $\eta$, and $t$ refers to the total period length.

## C  PROOFS OF MAJOR THEORETICAL RESULTS

**Proof of Lemma 1**. We introduce a procedure to verify if a selected set of nodes $V_s$ satisfies the required condition in PTIME. (1) The procedure first computes the temporal influence scores $\Phi(v_s, v)$ at each timestamp for each explanatory node $v_s \in V_s$ by definition. This is computable in PTIME. It then induces a set of temporal nodes $V_c$, edges $\mathbb{E}_s$ and $\mathbb{E}_c$ via $\delta$-reachability from $v_s$ to $v$. (2) It next verifies the counterfactual property of induced temporal edges, by invoking the inference process of $\mathcal{M}$, which is also in PTIME for mainstream TGNNs. If $V_s$ induces a set of adjacent edges $\mathbb{E}_\epsilon$ that are counterfactual set, then $V_s$ is a set of valid explanatory nodes. Putting these together, the verification process is in PTIME.

**Proof of Lemma 2** The hardness of the optimization problem, in its decision version, can be verified by a reduction from the maximum coverage problem (MCP), a known NP-hard problem. Given a universe set $\mathcal{U} = \{u_1, \ldots, u_n\}$, a collection of its subsets $\mathcal{S} = \{S_1, \ldots, S_m\}$, and an integer $k$, the

task is to select at most $k$ subsets from $\mathcal{S}$ to cover as many elements of $\mathcal{U}$ as possible. Given an instance $(\mathcal{U}, \mathcal{S}, k)$ of MCP, we construct in polynomial time an instance of lower-level optimization problem as follows. (1) Given the universal set $\mathcal{U}$, for each data point $u_i$ in $\mathcal{U}$, we set a distinct test nodes $v_{t_i}$ and include it into a test set $V_T$, with a fixed label $l$; (2) For each subset $S_i \subseteq \mathcal{U}$, we set a "group node" $v_i$, and for each data point $u_j \in S_i$, we add a temporal path $\rho_\delta$ from $v_i$ to $v_{t_j}$ to ensure $v_i$ $\delta$-reaches $v_{t_j}$. If $u_j \notin S_i$, then $v_i$ does not $\delta$-reaches $v_{t_j}$. (3) We choose a simple TGNN that contains a single input layer and a constant output layer that directly maps input nodes with constant features to a fixed label $l$, hence each group node $v_i$ is an explanatory nodes for the nodes in $V_T$ that can be $\delta$-reached (and temporally influenced) by $v_i$. We may then verify that there is a $k$-set of explanatory nodes that "cover" targeted set $V_T$, if and only if the corresponding $k$ subsets that can cover $\mathcal{U}$ at a threshold. As MCP is NP-hard, the hardness of low-level problem follows.

**Proof of Lemma 3**. Algorithm TemGX is a $(1 - \frac{1}{e})$-approximation for the instance-level optimization problem, *i.e.*, $\Phi(V_s, v) \geq (1 - \frac{1}{e})\Phi(V_s^O, v)$. We show this result by proving that the problem essentially aims to maximize a monotone submodular function under a cardinality constraint, a setting where the greedy algorithm guarantees a $(1 - \frac{1}{e})$ approximation (Qian et al., 2017).

Consider the candidate set $\mathcal{C}$ for the target output $\mathcal{M}(G_t, v)$ and let $\Phi(V_s, v)$ denote the temporal influence of any $V_s \subseteq \mathcal{C}$ on $\mathcal{M}(G_t, v)$. Let

$$V_s^O = \arg \max_{V_s \subseteq \mathcal{C}, |V_s| \leq k} \Phi(V_s, v).$$

The TGNN model is fully trained and kept fixed, and the input temporal graph window $G_t$ is also fixed during the explanation phase. All influence and resistance scores $\mathrm{inf}(v_s, v, t')$ and $\mathrm{trd}(v_s, v, t')$ are evaluated once on $G_t$ using the trained model and reused throughout the greedy procedure; no model retraining or graph updates are performed while the subset $V_s$ grows. Under this evaluation strategy the value of each pairwise score is constant during selection.

By the definitions in Section 3,

$$\Phi(V_s, v) = \frac{1}{\delta} \sum_{v_s \in V_s} \sum_{t'=t-\delta+1}^{t-1} \mathrm{inf}(v_s, v, t') \, g(\mathrm{trd}(v_s, v, t')) \, \exp(-\lambda(t - t')).$$

Define the singleton contribution

$$\phi(v_s, v) = \frac{1}{\delta} \sum_{t'=t-\delta+1}^{t-1} \mathrm{inf}(v_s, v, t') \, g(\mathrm{trd}(v_s, v, t')) \, \exp(-\lambda(t - t')),$$

so that $\Phi(V_s, v) = \sum_{v_s \in V_s} \phi(v_s, v)$.

**Monotonicity.** Because every term is non–negative, for any $V_s \subseteq \mathcal{C}$ and $v_s \in \mathcal{C} \setminus V_s$,

$$\Phi(V_s \cup \{v_s\}, v) - \Phi(V_s, v) = \phi(v_s, v) \geq 0.$$

**Submodularity.** Let $A \subseteq B \subseteq \mathcal{C}$ and $v_s \in \mathcal{C} \setminus B$. Since $\phi(v_s, v)$ is independent of the current subset,

$$\Phi(A \cup \{v_s\}, v) - \Phi(A, v) \geq \Phi(B \cup \{v_s\}, v) - \Phi(B, v),$$

Notice from the definition that the marginal gain introduced by adding node $v_s$ is independent of the nodes already present in the set. Specifically, the contribution of node $v_s$ is given explicitly by $\frac{1}{\delta} \sum_{t'=t-\delta+1}^{t-1} \mathrm{inf}(v_s, v, t') \cdot g(\mathrm{trd}(v_s, v, t')) \cdot \exp(-\lambda(t - t'))$, which is unaffected by the presence or absence of other nodes in the selected set. Consequently, the marginal gain is constant with respect to the subset context, trivially fulfilling the diminishing returns condition required for submodularity. Therefore, $\Phi(\cdot)$ is submodular over subsets of $\mathcal{C}$.

Because $\Phi(\cdot)$ is monotone submodular applying the standard greedy algorithm under the cardinality constraint $|V_s| \leq k$ yields a $(1 - \frac{1}{e})$ approximation guarantee. Therefore the set $V_s$ returned by TemGX satisfies

$$\Phi(V_s, v) \geq (1 - \frac{1}{e})\Phi(V_s^O, v),$$

The above analysis completes the proof.

**Performance analysis of** TemGX **for multiple target outputs.** By reducing the problem to maximum coverage problem (Vazirani, 2013), TemGX ensures the same guarantee of $1 - \frac{1}{e}$ as its counterpart for single targeted TGNN output, and can be implemented with the same worst-case time cost. To see this, we show that the algorithm only needs to invoke the single-test node counterpart (Algorithm 1) multiple types, and consistently apply a greedy selection strategy, that essentially solves a *maximum weighted coverage problem*.

The maximum weighted coverage problem takes as input a collection $\mathcal{S}$ of sets $\{S_1, \ldots S_n\}$, and aims to find a subset $\mathcal{S}' \subset \mathcal{S}$ of size $k$, such that the total weight of the elements in $\bigcup_{S \in \mathcal{S}'} S$ is maximized. Given graph $G$ as a single snapshot with a set of target nodes $V_T$ and their outputs, we construct the following reduction. (1) We set $\mathcal{S}$ to be $V_T$. (2) For each node $v$ in $G$, we invoke the verification procedure to check if any of its adjacent edge is a counterfactual edge for any target node $v_j \in V_T$; if so, add the node $v$ to a set of explanatory nodes $V_{\epsilon_j}$ corresponding to the target node $v_j$. We say $v$ "covers" $v_j$ in this case. This process takes at most $|V|$ verifications. (3) We then construct a universal explanatory node set $V_\epsilon$, which contains explanatory nodes that at least cover one target node in $V_T$. For each explanatory node $v_\epsilon \in V_\epsilon$, We initialize a set of subsets of nodes $\mathcal{S}$, such that each subset $S \in \mathcal{S}$ contains the test nodes covered by $v_\epsilon$. (4) We set the weights of each element as the temporal influence score $\Phi(v_\epsilon, v)$, for each pair of explanatory node $v_\epsilon$ and the target node $v$ it covers. We may then verify that the solution of the maximum weighted coverage problem, as a subset $S' \subset \mathcal{S}$, can be converted to a set of selected explanatory nodes that cover exactly a set $S$ of targeted nodes in $S'$. This provides a solution for node selection with multiple outputs.

The optimality guarantee follows from the greedy strategy applied for solving maximum weighted coverage problem, which is ensured by solving the problem of maximization of submodular functions with a cardinality constraint. As the sum of submodular functions remain to be a monotone submodular function, we only need to extend algorithm TemGX for $V_T$, which (1) for each node $v$ in $G$, verifies if each of its edges is in a counterfactual set, for every node in $V_T$, in a single batch, to decide the many-to-many coverage relation; and (2) initializes an instance of a maximum weighted coverage problem, and apply consistently the greedy selection strategy to obtain the results.

The overall cost is in $O(|V_T|t \times k \log k + |N_L(V_T)|T_I)$. The worst case occurs when each node in $V_T$ has a distinct set of explanatory nodes that are disjoint with any counterpart of the rest of the nodes in $V_T$.

# D    DETAILS OF DATASETS

**Datasets**. We use six real-world temporal graphs below (summarized in Table 2). (1) METR-LA (Li et al., 2018) and PEMS-BAY (Li et al., 2018) are transportation networks with traffic speeds in Los Angeles and Bay Areas, respectively, where nodes are locations and edges represent road segments. (2) Wiki (Kumar et al., 2019) includes one month of Wikipedia edits, with nodes as editors and pages and edges as timestamped posting requests. Edge features are 172-dimensional vectors from edited texts. (3) UCIM (Kunegis, 2013) is a temporal social interaction network of time-stamped user messages spanning 196 days with a high ratio of unique edges. (4) Multihost (Gao et al., 2021) is a cybersecurity dataset capturing multi-step attack scenarios, where nodes represent network hosts, processes, and files, and edges denote attack propagation events including process executions and network connections. (5) Elliptic++ (Elmougy & Liu, 2023) is a dynamic Bitcoin transaction dataset, where nodes correspond to wallet addresses (transactions) and temporal edges represent timestamped cryptocurrency transfers, with the task of classifying illicit transactions.

# E    EXPERIMENTAL SETTINGS

**Set up**. All methods and tests are implemented in Python 3.10.12 by PyTorch Geometric. Tests are deployed on a HPC cluster with Tesla V100-SXM2-32GB GPU (CUDA ver. 12.2), an NVIDIA driver (ver. 535.54.03), and a dual Intel Xeon Gold 6226 CPU with 24 cores at 3.7 GHz, managed by Slurm. All tests were conducted 5 independent times using different random seeds, and the average results are reported.

**Baseline Implementation and Parameter Alignment**

Table 2: Summary of datasets. snapshot: the average snapshot size in (# nodes, # edges); # feature: the average number of features per node/edge; period: the total time $t$, *i.e.,* number of snapshots in $\mathcal{G}_t$; $\delta$: window size; task: the task a TGNN is trained for.

| Dataset | snapshot | #feature | (period,$\delta$) | task |
|---------|----------|----------|-------------------|------|
| METR-LA | (207,1515) | 1 | (34272,12) | regression |
| PEMS-BAY | (325,2369) | 1 | (52128,12) | regression |
| Wiki | (9227,157474) | 172 | (731,14) | link prediction |
| UCIM | (1899,59835) | 1 | (196,12) | link prediction |
| Multihost | (6457,252457) | 8 | (66,4) | classification |
| Elliptic++ | (5156,4784) | 166 | (49,6) | classification |

All baseline explainers were implemented using their official repositories, and all experiments were conducted under identical backbone (TGN,TGAT), temporal reachability ($\delta = 12$), neighborhood depth ($L = 2$), and explanation budget ($k = 50$). (1)**TemGX** with $\delta = 12$, $L = 2$, $k = 50$. (2)**CoDy** (Qu et al.) with sample_size=50, candidates_size=128, and $\alpha = \frac{2}{3}$ using identical $L = 2$. We use CoDy-Spatio-Temporal, a recommanded variant (Qu et al.) due to its more comprehensive outputs compare with other variants in the CoDy family (which has also been verified by our tests). (3)**TGNNExplainer** (Xia et al., 2022) trained for 200 epochs with learning rate $10^{-3}$ and batch size 32, selecting top-$k = 50$ most influential temporal edges per target link. (4)**TempME** (Chen & Ying, 2023) with learning rate $1 \times 10^{-3}$, contrastive loss weight $\lambda_c = 1.0$, dynamic loss weight $\lambda_d = 0.5$ and fixed mask sparsity threshold corresponding to top-50 edges. (5)For regression datasets (METR-LA, PEMS-BAY), we maintain $k = 50$ to ensure comparability of temporal influence fidelity. Additional ablations in Appendix I.1 report results across $k \in \{10, 20, 30, 40, 50\}$, confirming consistent ranking and robustness.

**Evaluation Metrics**. For a fair comparison with (Xia et al., 2022; Chen & Ying, 2023), we use three common measures below. (1) The **fidelity** Fid measures the difference between the original and new prediction probabilities. Consider an induced explanatory temporal subgraph $\mathcal{G}_\epsilon$ with temporal edges $\mathbb{E}_\epsilon$, of an explainer. (a) For TGNNExplainer and TempME, $\mathbb{E}_s$ refers to a set of critical events (edges) as explanations. (b) For link prediction, we compare TemGX, TGNNExplainer, and TempME. Follow the convention (Xia et al., 2022) that measure the difference over the learned logits $Z_\epsilon = f_D(f_E(\mathcal{G}_\epsilon))$, we define $\text{Fid}(\mathcal{G}_\epsilon, V_T) = \mathbb{I}(Y_T = 1) \cdot (Z_\epsilon[V_T] - (Z[V_T]) + \mathbb{I}(Y_T = 0) \cdot (Z_\epsilon[V_T] - Z[V_T])$, where $\mathbb{I}(\cdot)$ is the indicator function, $Y_T$ is the true label of $V_T$, $Z[V_T]$ and $Z_\epsilon[V_T]$ represent the logits of $V_T$ on input graph $\mathcal{G}_t$ and $\mathcal{G}_\epsilon$, respectively. (c) For node regression tasks, TGNNs output predicted values rather than links, hence with their source codes, TempME and TGNNExplainer are not directly applicable. We compare TemGX and TGVex, and define fidelity as: $\text{Fid}(\mathcal{M}(\mathcal{G}_s)[V_T]) = 1 - \left( \frac{|\mathcal{M}(\mathcal{G}_s)[V_T] - \mathcal{M}(\mathcal{G}^t)[V_T]|}{|\mathcal{M}(\mathcal{G}^t)[V_T]|} \right)$ where $\mathcal{M}(\mathcal{G}_s)[V_T]$ and $\mathcal{M}(\mathcal{G}^t)[V_T]$ are the model predictions for node $V_T$ on the explanatory temporal subgraph $\mathcal{G}_\epsilon$ and the full graph $\mathcal{G}^t$, respectively. (2) The **sparsity** assesses the conciseness of the explanations. We normalize the sparsity as $\frac{\mathbb{E}_\epsilon}{|\mathbb{E}^L(V_T)|}$, where $\mathbb{E}^L(V_T)$ refers to the temporal edges within $L$-hop neighbors of the nodes in targeted nodes $V_T$ in the temporal graph $\mathcal{G}_T$, for $L$-layered TGNNs. (3) We evaluate performance using the **fidelity-sparsity curve** (FSC) and calculate the **Area Under the Fidelity-Sparsity Curve** (AUFSC), consistently following the baseline TGNN explainers. The higher, the better (Xia et al., 2022; Chen & Ying, 2023).

# F STUDY OF ABLATION ON UCIM+TGN

To evaluate the contribution of each component and the sensitivity of critical parameters, we conduct a series of ablation experiments on the UCIM dataset with a TGN backbone. These studies quantify the effect of temporal influence modeling, temporal resistance distance (TRD), and temporal decay on both the fidelity and sparsity of the generated explanations.

**Component analysis.** We first compare four model variants: (1) the full TemGX model, (2) TemGX without temporal decay, (3) TemGX without TRD, and (4) TemGX without ICM influence. The results show that removing any of these components leads to a clear drop in fidelity, with the absence of ICM producing the most significant degradation. For example, at the highest sparsity level the fidelity of the full model reaches **0.468**, whereas removing TRD reduces fidelity to 0.388 and re-

Table 3: Component ablation on UCIM+TGN. Fidelity is reported at the highest sparsity level.

| Variant | Fidelity ↑ | Relative Drop |
|---|---|---|
| Full TemGX | **0.468** | – |
| No Time Decay | 0.400 | −14.5% |
| No TRD | 0.388 | −17.1% |
| No ICM | 0.308 | −34.2% |

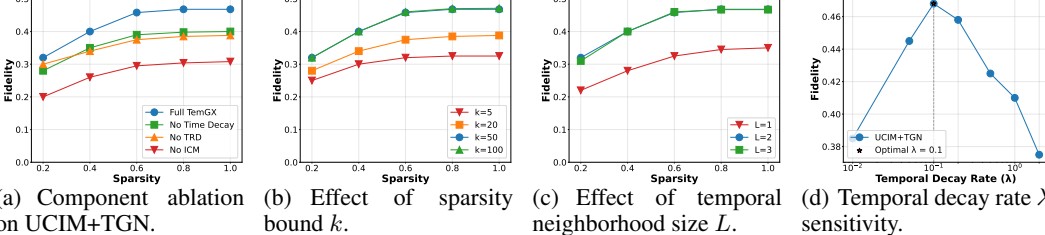

(a) Component ablation on UCIM+TGN.    (b) Effect of sparsity bound $k$.    (c) Effect of temporal neighborhood size $L$.    (d) Temporal decay rate $\lambda$ sensitivity.

Figure 6: Ablation results on **UCIM+TGN**. (a) compares component settings, (b) studies the impact of sparsity bound $k$, (c) evaluates the influence of the $L$-hop temporal neighborhood, and (d) analyzes temporal decay rate $\lambda$ sensitivity.

moving ICM reduces fidelity to 0.308. These observations confirm that both temporal influence computation and structural resistance are important for achieving high–quality explanations.

**Parameter sensitivity.** We next study the influence of three key hyperparameters: the sparsity bound $k$, the $L$–hop temporal neighborhood size, and the temporal decay rate $\lambda$. Increasing $k$ allows larger explanation subgraphs and steadily improves fidelity, from 0.325 at $k = 5$ to **0.468** at $k = 50$. Fidelity rises sharply when enlarging the neighborhood from $L = 1$ to $L = 2$, but beyond $L = 2$, indicating that two–hop temporal neighbors capture most of the informative context.

For the decay parameter $\lambda$, we conduct a sensitivity analysis across values ranging from 0.01 to 1.0 as shown in the bottom right of Figure 6. The results reveal a inverted-U shaped relationship between $\lambda$ and fidelity. Very small values ($\lambda = 0.01$) yield suboptimal performance (fidelity 0.385) as they fail to capture distant temporal information. Conversely, large values ($\lambda = 1.0$) also degrade performance (fidelity 0.375) by over-emphasizing recent snapshots and losing long-range temporal dependencies. The optimal value $\lambda = 0.1$ achieves peak performance (fidelity **0.468**), effectively balancing recent and historical temporal influence. This systematic parameter sweep validates our choice of $\lambda = 0.1$ and addresses concerns about arbitrary parameter selection.

Overall, these results demonstrate that the proposed temporal influence computation, temporal resistance distance, and decay mechanism are all essential for achieving high fidelity and compact explanations. The sensitivity analyses further guide practical parameter choices: a 2–hop neighborhood ($L = 2$), a moderate sparsity bound ($k = 50$), and a decay rate around $\lambda = 0.1$ consistently deliver a favorable tradeoff between interpretability and predictive faithfulness on UCIM+TGN.

## G STUDY OF ABLATION ON METR-LA+STGCN

To evaluate the generalizability of our approach across different domains and tasks, we conduct ablation experiments on the METR-LA dataset with an STGCN backbone for spatio-temporal regression. These studies examine how component contributions and parameter sensitivities differ between social network analysis (UCIM+TGN) and traffic prediction (METR-LA+STGCN), providing insights into domain-specific explanation requirements.

**Component analysis.** We compare the same four model variants: (1) the full TemGX model, (2) TemGX without temporal decay, (3) TemGX without TRD, and (4) TemGX without ICM influence. Interestingly, the component importance ranking differs significantly from the classification task on UCIM. For the traffic regression task, removing temporal decay produces the most severe performance degradation, with fidelity dropping from **0.471** to 0.393 (a 16.6% decrease). TRD removal

Table 4: Ablation study on METR-LA+STGCN. Fidelity is reported at the highest sparsity level.

| Variant | Fidelity ↑ | Relative Drop |
|---|---|---|
| Full TemGX | **0.471** | – |
| No Time Decay | 0.393 | −16.6% |
| No TRD | 0.413 | −12.3% |
| No ICM | 0.443 | −5.9% |

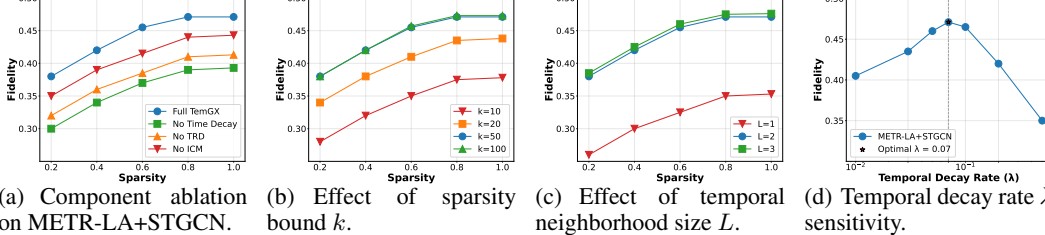

(a) Component ablation on METR-LA+STGCN.
(b) Effect of sparsity bound $k$.
(c) Effect of temporal neighborhood size $L$.
(d) Temporal decay rate $\lambda$ sensitivity.

Figure 7: Ablation results on **METR-LA+STGCN**. (a) compares component settings, (b) studies the impact of sparsity bound $k$, (c) evaluates the influence of the $L$-hop temporal neighborhood, and (d) analyzes temporal decay rate $\lambda$ sensitivity for traffic regression.

causes a 12.3% performance drop (fidelity 0.413), while ICM removal has the smallest impact with only 5.9% degradation (fidelity 0.443).

**Parameter sensitivity.** We examine the same hyperparameters, but observe distinct optimal values reflecting traffic data characteristics. The impact of the sparsity bound $k$ exhibits similar trend, consistent with the performance plateauing at $k = 50$ (fidelity 0.471) and minimal improvement at $k = 100$ (fidelity 0.473). The $L$-hop neighborhood analysis reveals that the results at $L = 3$ slightly outperforms its counterpart at $L = 2$ in traffic networks (fidelity 0.476 vs 0.471), capturing longer-range spatial propagation of congestion effects along interconnected road segments. Most notably, the temporal decay rate $\lambda$ requires different discounting strategies for traffic data, with optimal performance at $\lambda = 0.07$ compared to $\lambda = 0.1$ for social interactions. This reflects the finer temporal granularity in traffic patterns (5-minute intervals), where maintaining more historical information is crucial for capturing traffic cycles and rush-hour dynamics.

The domain-specific differences reveal important insights about temporal graph explanation. In traffic networks, temporal patterns (captured by decay mechanisms) are even more dominant than in social networks, with a 16.6% performance drop compared to 14.5% in UCIM, as traffic flow follows highly predictable temporal cycles with rush-hour patterns. The temporal resistance distance remains important but shows increased significance (12.3% vs 17.1% in UCIM), reflecting the continuous spatial propagation nature of traffic congestion compared to discrete social connections. Notably, ICM influence shows dramatically reduced importance in traffic regression (5.9% vs 34.2% in social classification), suggesting that influence maximization concepts are less relevant for physical flow prediction than social interaction modeling. These findings demonstrate that while TemGX maintains some consistent parameter optimization strategies ($k = 50$), both the relative component importance and temporal decay requirements must be carefully adapted to domain-specific characteristics. The sensitivity analyses further guide practical parameter choices for traffic applications: a 2-hop neighborhood ($L = 2$) to capture extended spatial propagation, a moderate sparsity bound ($k = 50$), and a finer temporal decay rate ($\lambda = 0.07$) that preserves more historical information for spatio-temporal regression on METR-LA+STGCN.

## H  TemGX IN THE WILD: APPLICATIONS

**Case I: Interpretable Illicit Account Detection in Bitcoin Transactions**. The Bitcoin transactions dataset is derived from Elliptic++ (Elmougy & Liu, 2023), where nodes are wallet addresses and temporal edges are timestamped transactions. We train **TGN** and **TGAT** for illicit-address classification and apply TemGX for instance-level explanations.

*Parameter settings.* We set the hop size $L = 2$, temporal decay $\lambda = 0.1$, and the explanation size $k = 5$ per window. Each window uses a distinct test node.

Results. Table 5 showcases a set of explanatory subgraphs reported by TemGX. The Red nodes denote the explanatory set $V_s$, blue nodes are connecting nodes $V_c$, and the green node is the target host $V_t$. and the corresponding ground truth. We report the two summarized patterns. We found that these two patterns are highly consistent with the verified money laundering patterns in Bitcoin transaction: *Peel Chain* and *Spindle*. Moreover, TemGX can capture how the transactions "evolve" from one strategy to another.

For example, in the time window $W_1$ in Table 5, TemGX generates an explanatory subgraph for the target node $4827$ (classified by TGN as an illicit account) that clearly reveals a real Bitcoin "peel chain" money-laundering pattern. The most influential explanatory nodes include $1247$, $2103$, $892$, $4785$, and $921$: among them, $2103$ acts as an upstream fund aggregation hub that frequently receives large external transfers; $1247$ and $892$ exhibit high-frequency micro-transactions and frequent address changes, serving as typical relay accounts; and $4785$ and $921$ provide supplementary fund inputs to $4827$ within a short time span. The main transaction paths appear as $2103 \rightarrow 1247 \rightarrow 4827$, $4785 \rightarrow 892 \rightarrow 4827$, and a direct transfer $921 \rightarrow 4827$, forming a multi-source peel-chain structure.

Leveraging the $\delta$-reachability constraint, TemGX ensures that these nodes remain temporally connected to $4827$, thereby isolating the most time relevant upstream fund sources likely to influence the TGNN model's prediction. The resulting explanation not only covers all key accounts and transaction paths of the peel chain, but also strictly respects temporal constraints, providing verifiable and practically meaningful evidence for classifying $4827$ as an illicit account. Unlike TemGX, the subgraphs generated by CoDy and TGNNExplainer lack this temporally coherent, $\delta$-reachable structure and fail to capture the temporal peel-chain transactions.

## Case II: Explainable Multi-step Cyberattack Detection

The Cyberattack dataset (Gao et al., 2021) is constructed from system audit logs, where nodes represent hosts, processes, or files and temporal edges record timestamped events. We train **TGN** and **TGAT** for host compromise classification and apply TemGX for instance–level explanations.

*Parameter settings.* To capture long–range attack chains, we set the hop size $L = 3$, temporal decay $\lambda = 0.1$, and explanation size $k = 4$ per window. Distinct target hosts are sampled for each window.

Results. Table 6 demonstrates that TemGX identifies critical multi-stage cyber attack pathways across four representative time windows, with the discovered explanatory subgraphs (red nodes) and connecting nodes (blue nodes) capturing the temporal progression from initial compromise through privilege escalation to data exfiltration that domain experts recognize as realistic attack scenarios.

For example,in time window $W_4$ in Table 6, TemGX generates an explanation subgraph for the target node 235 (classified by the TGAT as a compromised core server srv235) that reconstructs the multi-stage attack chain from initial intrusion to data exfiltration. The key explanatory nodes include srv294, srv900, proc787, relay173, and relay442. srv294 records the first cross-domain login attempt following external scanning srv900 subsequently escalates privileges to obtain higher access rights; proc787 executes repeated MD5 hash operations in a short period, corresponding to the attacker preparing data packaging; relay173 acts as a cross-subnet relay, forwarding multiple files containing sensitive credentials to relay442 within 10 minutes; finally, relay442 establishes a high-privilege session with srv235 and triggers an abnormal read of finance_report.zip, which aligns with known data-exfiltration behavior.

TemGX's temporal $\delta$ reachability together with temporal resistance distance (TRD) accurately isolates nodes that are both temporally and structurally closest to srv235, ensuring that each attack path (e.g., srv294→srv900→proc787→relay173→relay442→srv235) respects the real attack timeline and causal order. In contrast, TGNNExplainer and CoDy contain only a few isolated nodes. This structurally consistent explanation not only provides direct evidence for the TGNN prediction that srv235 is compromised, but also enables security analysts to reproduce the attack scenario and design targeted defense strategies.

Table 5: Explanations by TemGX on Elliptic++ (Elmougy & Liu, 2023).

| Time Window | Target Node ID | TGN (TemGX) | TGAT (TemGX) | Ground Truth | Description |
|---|---|---|---|---|---|
| $W_1$ | 4827 |  |  |  | Peel Chain. |
| $W_2$ | 4123 |  |  |  | Peel Chain. |
| $W_3$ | 3291 |  |  |  | Spindle. |
| $W_4$ | 3011 |  |  |  | Spindle. |

Table 6: Explanations by TemGX on the Cyberattack (Gao et al., 2021).

| Time Window | Target Host ID | TGN (TemGX) | TGAT (TemGX) | Ground Truth | Description |
|---|---|---|---|---|---|
| $W_1$ | 664 |  |  |  | Initial compromise chain to the target host. |
| $W_2$ | 702 |  |  |  | Privilege escalation following compromise. |
| $W_3$ | 233 |  |  |  | Data exfiltration via covert/relay steps. |
| $W_4$ | 235 |  |  |  | Data exfiltration via covert/relay steps. |

**Case III: Comparison of explanation structures**. We analyze the structural characteristics of explanations generated by different temporal graph explainers on the Wiki datasets (Table 7), revealing key methodological differences:

*Temporal Cohesiveness.* In the Wiki link prediction task, each target edge $(u, p, t)$ represents that user $u$ edits page $p$ at time $t$. CoDy relies on Monte Carlo sampling and backtracking, frequently combining events from disjoint time periods that are not $\delta$-reachable, producing temporally fragmented subgraphs. TempME tends to select local motifs on each side of the target edge, resulting in two separated clusters with no temporal bridge. TemGX, in contrast, enforces $\delta$-reachability so that only events within the time window and connected to $(u, p, t)$ are retained. The resulting explanations form *time-respecting paths*, such as "co-edit $\rightarrow$ short gap $\rightarrow$ repeated co-edit $\rightarrow$ target edit," avoiding the disconnected event islands seen in the baselines.

*Queryable Capabilities.* TGNNExplainer, CoDy, and TempME output a single static subgraph that cannot be further queried. Analysts cannot directly ask questions such as "within a $\delta$-window, which group of co-editing users jointly influence the target edge?" or "what is the impact of removing cross-page edits within the last hour?". TemGX outputs a queryable temporal subgraph: under the $\delta$-reachability constraint, analysts can issue temporal pattern queries. Moreover, TemGX supports probabilistic summarization via a Dynamic Bayesian Network (DBN), condensing frequent temporal patterns (e.g., "near-time co-edit $\rightarrow$ target edit") into reusable transition probabilities for analysis.

**Computational Considerations**. TGNNExplainer requires additional training of a navigator incurring high training and explanation costs. CoDy depends on large-scale Monte Carlo candidate sampling with iterative backtracking, which is computationally expensive. TempME performs random-walk motif matching, and its inference time grows quickly with motif complexity or candidate size. TemGX is training-free and uses a one-pass greedy selection guided by ICM and TRD scores. The $\delta$-reachability prunes temporally irrelevant events and keeps the search space within a time window around the target, ensuring stable runtime and memory usage even on large temporal graphs.

Overall, for Wiki's link prediction task, TGNNExplainer's navigator produces temporally disconnected explanations, CoDy's sampling yields cross-window fragments, and TempME fails to capture continuous time decay. TemGX ensures temporal cohesiveness, integrates TRD with temporal decay modeling, and supports queryable probabilistic analysis, delivering explanation subgraphs with temporal consistency and practical interpretability.

Table 7: Comparison of Explanation Subgraphs on Wiki Dataset (Fixed $k = 5$).

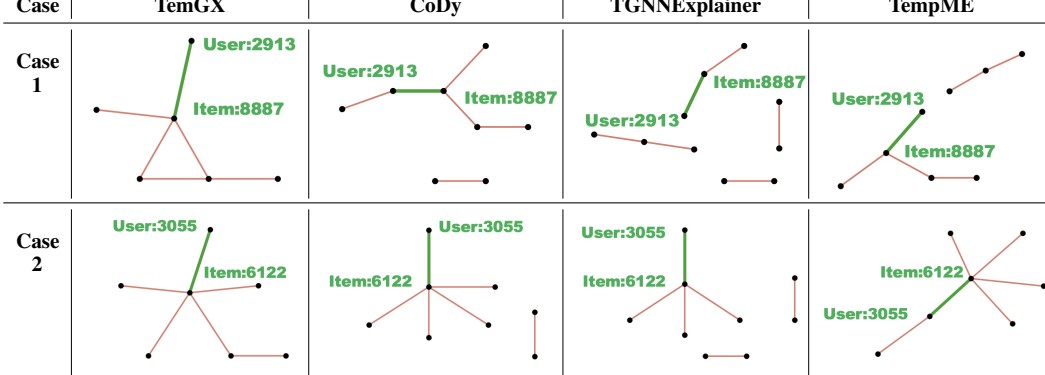

# I STUDY OF QUERYING TEMPORAL EXPLANATIONS

**Exp-4: Querying Temporal Explanations**. We next showcase how TemGX supports queryable explanation for real-world TGNN use cases.

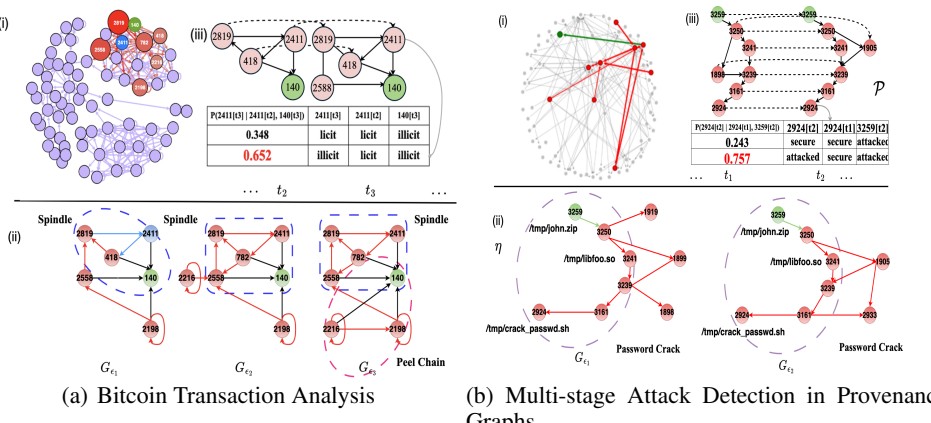

(a) Bitcoin Transaction Analysis

(b) Multi-stage Attack Detection in Provenance Graphs

Figure 8: Querying Temporal Explanations Case Analysis

*Clarifying Suspicious Transactions.* Fig. 5 (i) shows a fraction of the Bitcoin network around node 140 (an "illicit" IP detected by a TGN). Fig. 5 (ii) illustrates three temporal explanatory subgraphs $\mathcal{G}_{\epsilon_1}$-$\mathcal{G}_{\epsilon_3}$ generated by TemGX for node 140. To demonstrate instance-level querying capabilities, consider temporal pattern queries $Q_\eta = (V_Q, E_Q, \sigma, \delta')$ posed on the discovered explanations. For example, a "Spindle" pattern query with nodes $\{418, 2819, 2411\}$ can be efficiently matched against $\mathcal{G}_{\epsilon_1}$ at the first timestamp through TemGX algorithm. Similarly, querying for "Peel Chain" patterns involving node 2216 successfully retrieves matches from $\mathcal{G}_{\epsilon_3}$, highlighting how explanatory structures evolve over time.The querying interface allows users to search for specific temporal dependencies: "Find all explanatory subgraphs where node 418 transitions to 782 as Spindle participants" returns $\mathcal{G}_{\epsilon_2}$ with updated participants $\{782, 2558, 2411, 2819\}$. Such queries enable the investigation of suspicious transactions that are well-grounded by money-laundry patterns, and track their temporal evolution across different laundering strategies via DBN inference.

Fig. 8 (iii) illustrates a fraction of summary explanation $\mathcal{P}$, referring to explanatory IPs and their temporal states over time, which allows analysts to query cross-time dependencies that standard static models would miss. For example, an inference query that asks "*what is the likelihood of an 'licit' account 2411 to be 'illicit'?*" returns that 2411 has a 63.10% chance to remain licit and 36.90% chance to become illicit at time $t_2$, yet a higher chance 65.20% at time $t_3$. Indeed, 2411 is under more substantial influence from illicit IP 140, grounded by Spindle and Peel Chain matches in $\mathcal{G}_{\epsilon_3}$ at instance level.

*Multi-stage Cyber Attack Analysis.* Fig. 5 (i) illustrates a multi-stage cyber attack scenario from the Multihost dataset, where TemGX enables querying of malicious activities across network hosts.For instance, querying "Find attack paths involving gather_password.sh followed by crack_passwd.sh" efficiently retrieves the temporal dependency chain from host1 to host2 across the discovered subgraphs in Fig. 5 (ii).

Security analysts can pose more complex temporal queries such as "Identify hosts where Shellshock vulnerability leads to payload download within $\delta' = 2$ time steps", which matches the attack progression involving libfoo.so download on host2. Similarly, data exfiltration queries targeting "MD5 hash operations on john.zip" can be processed to trace the complete attack kill chain.Such instance-level querying enables security teams to investigate attack patterns, validate incident response procedures, and identify similar attack vectors across different temporal windows, transforming explanatory subgraphs into actionable threat intelligence.

Fig. 8 (iii) illustrates a fraction of summary explanation $\mathcal{P}$. An inference query that asks "*what is the likelihood of host 2924 being attacked if host 3259 is attacked?*" posed on TemGX returns that 2924 has only a 5% risk when host 3259 is secure, yet when host 3259 is attacked, this probability increases to 32.1% at the second timestamp and further escalates to 75.7% during the lateral movement phase. This demonstrates TemGX's ability to provide grounded explanations for complex multi-step cyber attack analysis.

## J  FIXED EXPLANATION SIZE ANALYSIS

We conduct additional experiments evaluating all methods at fixed explanation sizes $k \in \{10, 20, 30, 40, 50\}$. This controlled setting ensures that performance differences reflect edge selection quality rather than the number of edges used. We focus on UCIM and Wiki datasets with TGN backbone. At $k = 50$, this corresponds to fidelity values of 0.468 and 0.262, matching Table 1 configurations.

*Fidelity Analysis.* Figure 9 (a-b) shows that TemGX consistently outperforms baselines across all $k$ values on both datasets. On UCIM, TemGX achieves 18.8% higher fidelity than CoDy at $k = 50$ (0.468 vs 0.394), with the gap widening from 6.8% at $k = 10$. Critically, while all methods show improvement with larger $k$, CoDy shows 43.3% improvement from $k = 10$ to $k = 50$, whereas TemGX maintains steady growth (58.6% improvement). This validates our contribution: TemGX's ICM×TRD identifies structurally critical edges even when temporally distant.

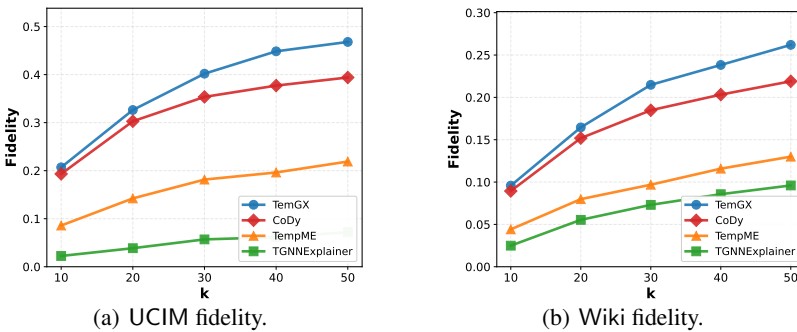

(a) UCIM fidelity.        (b) Wiki fidelity.

Figure 9: Fixed explanation size comparison on UCIM and Wiki datasets with TGN backbone.

## K  FLIP-RATE ANALYSIS

We have added flip-rate analysis reporting the frequency of prediction changes after removing explanatory subgraphs across all datasets and backbones. For a target node $v_t$ with prediction $M(G_t, v_t) = y_{\text{pred}}$, we define flip-rate as: $flip-rate = \frac{|\{v_t \in V_T : M(G_t \setminus G_\epsilon, v_t) \neq y_{\text{pred}}\}|}{|V_T|}$ where $G_\epsilon$ is the discovered explanatory subgraph and $V_T$ is the set of test targets. For classification and link prediction tasks, "$\neq$" indicates different predicted class/link; for regression, "$\neq$" means $|M(G_t) - M(G_t \setminus G_\epsilon)| > 0.05 \times |M(G_t)|$ (i.e., more than 5% relative change).

*Results:* Table 8 presents flip-rate statistics across all experimental configurations. Flip-rates consistently range from 81–87% across all 12 dataset-backbone configurations, confirming counterfactual validity across diverse tasks and application domains. Removing the explanatory subgraphs changes the majority of predictions, demonstrating that TemGX identifies important temporal-structural subgraph rather than merely correlated features.

## L  TEMPORAL PARAMETER SELECTION AND THEORETICAL RELATIONSHIP

The $\delta$-reachability threshold and temporal decay rate $\lambda$ are theoretically coupled rather than independent. Following temporal kernel theory (Mei & Eisner, 2017; Trivedi et al., 2019), we suggest $\lambda = \alpha/\delta$, where $\alpha \in [0.8, 1.2]$ normalizes for dataset-specific temporal density. This ensures the effective time $(1/\lambda)$ aligns with the time window $\delta$, preventing overly rapid or slow decay. At the reachability boundary $(t = \delta)$, the temporal weight $e^{-\lambda\delta} = e^{-\alpha} \approx 0.37$–$0.45$, providing a meaningful decay consistent with TGNN kernels (Xu et al., 2019).

*Practical setting.* We choose $\delta$ based on domain-specific temporal scales—e.g., $\delta = 12$ for social graphs (UCIM, Wiki), $\delta = 12$ for traffic data (METR-LA, PEMS-BAY), and $\delta = 4$–$6$ for cybersecurity logs (Multihost). Given $\delta$, $\lambda$ is initialized as $1/\delta$ and slightly adjusted (e.g., $\lambda \approx 0.07$–$0.1$).

Table 8: Flip-Rate Analysis: Frequency of Prediction Changes After Removing Explanatory Subgraphs. Results shown for all dataset-backbone configurations.

| Dataset | Model | Task | Flip-rate | Avg Fidelity |
|---------|-------|------|-----------|--------------|
| UCIM | TGN | Link Prediction | 87.3% | 0.468 |
| | TGAT | Link Prediction | 85.6% | 0.415 |
| Wiki | TGN | Link Prediction | 81.2% | 0.262 |
| | TGAT | Link Prediction | 83.7% | 0.335 |
| METR-LA | STGCN | Regression | 84.6% | 0.471 |
| | DCRNN | Regression | 83.1% | 0.436 |
| PEMS-BAY | STGCN | Regression | 85.2% | 0.473 |
| | DCRNN | Regression | 83.9% | 0.441 |
| Multihost | TGN | Classification | 84.2% | 0.281 |
| | TGAT | Classification | 82.5% | 0.267 |
| Elliptic++ | TGN | Classification | 85.4% | 0.358 |
| | TGAT | Classification | 83.6% | 0.342 |

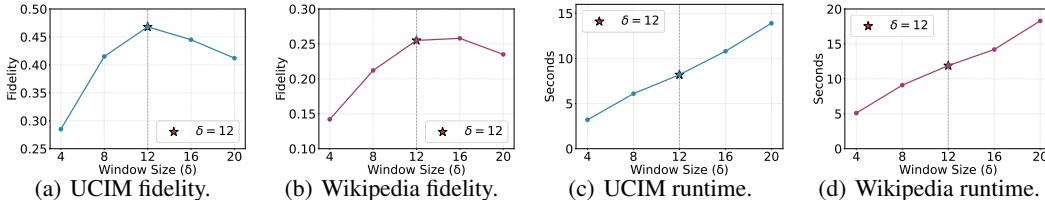

(a) UCIM fidelity.   (b) Wikipedia fidelity.   (c) UCIM runtime.   (d) Wikipedia runtime.

Figure 10: Window size sensitivity analysis on UCIM and Wiki with TGN backbone.

## M   WINDOW SIZE SENSITIVITY ANALYSIS

We analyze the impact of window size ($\delta$) on TemGX performance to validate our parameter selection. We evaluate TemGX on UCIM and Wiki datasets with varying $\delta \in \{4, 8, 12, 16, 20\}$ while keeping other parameters fixed ($k = 50$, $L = 2$, $|V_T| = 100$).

*Results and Analysis*: Figure 10 presents the sensitivity analysis results. We observe distinct patterns across datasets: UCIM **(Figures 10(a) and 10(c))**: Fidelity exhibits a inverted U-shape curve, achieving peak performance at $\delta = 12$ (fidelity=0.468). Smaller windows ($\delta < 12$) suffer from incomplete temporal coverage, missing long-range dependencies that contribute to TGNN predictions. For instance, $\delta = 4$ achieves only 0.285 fidelity (39% lower than optimal), as critical historical events beyond the 4-snapshot window are excluded. Conversely, larger windows ($\delta > 12$) introduce noise from distant, weakly relevant events, degrading fidelity to 0.412 at $\delta = 20$ (12% below peak). Runtime grows with $\delta$ due to early termination in our greedy selection, scaling from 3.2s at $\delta = 4$ to 13.8s at $\delta = 20$. Wiki **(Figures 10(b) and 10(d))**: Fidelity demonstrates a plateau region between $\delta = 12$ and $\delta = 16$ (0.255–0.258), indicating robustness to window size within this range. This plateau reflects Wikipedia's denser temporal graph structure, where editing patterns involve multiple interconnected snapshots. We select $\delta = 12$ as it provides optimal fidelity (0.255) with lower computational cost (11.8s) compared to $\delta = 16$ (14.2s), maintaining consistency with UCIM's optimal setting. Similar to UCIM, very small windows ($\delta = 4$: 0.143 fidelity) fail to capture sufficient temporal context, while excessively large windows ($\delta = 20$: 0.235 fidelity) accumulate noise.

## N   SCALABILITY ANALYSIS

To further validate the efficiency of TemGX under varying input scales, we conduct a scalability test with respect to (1) the number of explanatory edges $k$ and (2) the number of target nodes $|V_T|$. All

experiments are performed on the Wiki dataset with TGN backbone under the same hyperparameter configuration ($L = 2$, $\delta = 12$).

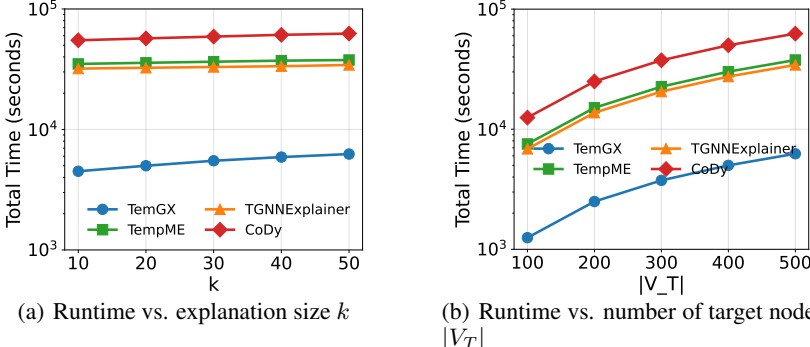

(a) Runtime vs. explanation size $k$

(b) Runtime vs. number of target nodes $|V_T|$

Figure 11: Scalability test of TemGX and baseline explainers. (a) shows runtime growth with respect to explanation size $k$, and (b) with respect to target node count $|V_T|$. Both are shown in log scale.

*Results and Analysis.* Figures 11(a) and 11(b) report the total computation time of TemGX, TGN-NExplainer, TempME, and CoDy. We observe that TemGX consistently achieves the lowest runtime across all settings. As $k$ increases from 10 to 50, TemGX increase from $2.5 \times 10^3$ to $6.3 \times 10^3$ seconds, while TGNNExplainer and TempME range between $3.4 \times 10^4$–$3.8 \times 10^4$ seconds, and CoDy grows to over $6.2 \times 10^4$ seconds due to Monte Carlo Tree search. Similarly, as $|V_T|$ increases from 100 to 500, TemGX's total runtime rises from $1.2 \times 10^3$ to $6.2 \times 10^3$ seconds, whereas TGNNExplainer and TempME increase from roughly $6.8 \times 10^3$–$3.8 \times 10^4$ seconds, and CoDy expands from $1.2 \times 10^4$ to $6.3 \times 10^4$ seconds. This scalability test verifies that TemGX's computational cost grows with local neighborhood size rather than global graph scale. The scalability advantage primarily arises from the greedy selection and verification mechanism that avoids redundant traversal over the temporal graph, in contrast to iterative retraining or exhaustive search required by model-specific explainers.

