# OpenReview forum: "Training-free Counterfactual Explanation for Temporal Graph Model Inference"
_ICLR.cc/2026/Conference — ICLR 2026 Poster_

### Official Review · Reviewer_Kj7f · 2025-10-22

**Soundness:** 3
**Presentation:** 2
**Contribution:** 3
**Rating:** 8
**Confidence:** 4

**Summary:**

The paper proposes an efficient explanation method for temporal graph neural networks. They introduce several engineering techniques and approximation algorithms to find the set of important input information that explains the model's decision.

**Strengths:**

I think this is a solid paper. The problem is well analyzed and the reasoning and technicalities of the method make sense to me. I also think the experimental results are thorough and carefully designed.

**Weaknesses:**

I expect general readers will have a hard time digesting this paper since it involves a lot of technicalities (from TGNN, explanation methods for it and some background in approximation algorithms), I would prefer if the authors can present the work in more intuitive manners.

**Questions:**

Previously, the work "On the Limit of Explaining Black-box Temporal Graph Neural Networks" (https://arxiv.org/pdf/2212.00952) pointed out that there exists certain information of TGNN that can never be faithfully recovered if only perturbation are used to examine TGNN. I am wondering how this work addresses the problem pointed out in the above paper?

---

> ### Author Response · Authors · 2025-11-21
>
> We thank the reviewer for the thoughtful assessment and constructive question. We appreciate the opportunity to clarify how TemGX addresses the fundamental limitations identified in Vu & Thai (2022).
>
> Vu & Thai (2022) identify three limits of perturbation-based temporal GNN explainers. TemGX addresses each through $\delta $-reachability ,multiplicative influential score and counterfactual verification.
>
> (i) Node perturbation cannot recover the temporal paths used by the TGNN.
>
> TemGX restricts candidates to the $\delta $-reachable L-hop neighborhood (Algorithm TemGX, line 7).This ensures that every selected node lies on a time-respecting path within the TGNN’s aggregation range, rather than being chosen solely because a perturbation changes the prediction.This mechanism reconstructs the temporal propagation structure that the TGNN relies on, which perturbation cannot guarantee. For example, TemGX successfully reconstructs the “spindle”-like path (2819 → 2411 → 418 → target), demonstrating faithful temporal path recovery.
>
> (ii) Edge perturbation cannot identify all nodes contributing to the prediction.
>
> TemGX ranks candidates using a multiplicative influential score (ICM × TRD × time-decay).ICM captures temporal influence propagation; TRD identifies nodes with high temporal-structural relevance across $\delta $-reachable window.This combination recovers nodes that contribute through feature influence, multi-path propagation, or cross-time connectivity, which edge-only perturbation may miss.
>
> (iii) Perturbing both nodes and edges does not reveal which components drive temporal aggregation.
>
> TemGX performs counterfactual verification (Algorithm TemGX, line 12–14).A candidate is retained only if removing the entire $\delta $-reachable explanatory subgraph changes the TGNN’s prediction.
> This avoids the failure mode identified by Vu & Thai, where uniform perturbation across time distorts the TGNN’s actual temporal aggregation behavior.
>
> We have added this discussion to Section 6 to clarify how TemGX addresses the issues highlighted by Vu & Thai (2022).

---

### Official Review · Reviewer_PEm2 · 2025-10-30

**Soundness:** 3
**Presentation:** 3
**Contribution:** 2
**Rating:** 6
**Confidence:** 3

**Summary:**

The paper proposes TemGX, a training-free, model-agnostic, and counterfactual explanation framework for TGNNs that: (1) efficiently identifies temporal explanatory subgraphs responsible for a model prediction. (2) propose a class of temporal explainability measures to verify these explanations. Experimental results show that TemGX outperforms other TGNN explainer methods in AUFSC, Fidelity and running time.

**Strengths:**

- The paper is well-written, and the case analyses provide good insights into the real-world applicability of the proposed framework.
- Experimental results show that TemGX outperforms other TGNN explainers on AUFSC, fidelity, as well as efficiency.
- The paper provides formal proofs that TemGX is a (1-1/e)-approximation, and it is in PTIME to verify the counterfactual properties for temporal subgraphs, which are meaningful contributions.
- The paper provides an analysis of the impact of different components (ICM, TRD, and temporal decay) of the temporal explainability measure on the fidelity score and shows that each component has a meaningful impact.

**Weaknesses:**

- It seems like the temporal explainability measure is a composition of established concepts, i.e. temporal effective resistance distance was previously proposed in (Zhu et al., 2024; Black et al., 2023), temporal decay was previously proposed in (Mei & Eisner, 2017), and temporal impact is an extension of the Independent Cascade Model. Can the authors clarify the novelty of this formulation?
- There is a lack of scalability analysis in the paper. It is unclear how the algorithm will scale to large temporal graphs. Even if the current algorithm is not scalable to large temporal graphs, the authors should still provide a theoretical or empirical scalability discussion.
- It seems like the algorithm is sensitive to the choice of the temporal decay rate λ. Does that mean we have to re-run hyperparameter sweeping for λ on each new dataset or task?

**Questions:**

- Please see Weaknesses section. Below are some extra questions:
- Can the author provide insights on how window size can impact TemGX performance?
- Temporal decay also seems to provide limited performance improvement compared to ICM and TRD. Can the authors provide insights on why this is the case.

---

> ### Author Response · Authors · 2025-11-21
>
> We thank the reviewer for the careful reading and the constructive feedback.
> ## W1
>
> The novelty of  TemGX does not lie in introducing ICM, resistance distance, or time decay individually, but in how these components are adapted to temporal TGNN explanation and integrated into one optimization objective.
>
> (1) Multiplicative integration adapted to TGNN counterfactuals:We introduce a single scoring function :$$ \Phi(E_s)=\text{ICM}(E_s)\times \text{TRD}(E_s)\times e^{-\lambda\Delta t} $$ Prior work applied these components separately or additively. Black et al. (2023) used effective resistance for static GNN analysis; Zhu et al. (2024) used resistance distance for graph rewiring; Mei & Eisner (2017) introduced time decay for point processes. Our contribution is to combine temporal influence, temporal structural distance, and time decay into one multiplicative score designed for TGNN counterfactual explanations. Prior work studied these components separately in different settings, but did not combine them for temporal counterfactual reasoning.
>
> (2) Temporal Resistance Distance: We define TRD as $\text{trd}(v_s, v, t’) = (Z_{t’}(v_s)-Z_t(v))^\top L_{t’}^\dagger (Z_{t’}(v_s)-Z_t(v))$. This formulation differs from the static resistance distance used by Black et al. (2023), where embeddings are compared at the same time.TRD compares embeddings across $\delta$-reachable time, capturing temporal propagation paths that TGNNs rely on.
>
> (3) Submodular optimization guarantee: The influential score $\Phi$ is submodular. Lemma 3 shows that maximizing $\Phi$ with a greedy strategy achieves a (1-1/e)-approximation. This provides a formal guarantee on explanation quality.
>
> (4) Component contribution verified by ablation:Our ablation results  (see Appendix F and G) show that the components contribute differently depending on the task: UCIM link prediction: removing ICM reduces fidelity by (−34.2%), removing time decay reduces fidelity by (−14.5%). METR-LA regression: removing time decay reduces fidelity by (−16.6%), removing ICM reduces fidelity by (−5.9%). These results show that the multiplicative interaction is necessary. A single component cannot perform well across tasks, and the combined form adapts to the prediction objective naturally.
>
> (5) Pluggable structure: The framework allows: $$\Phi(E_s)=f_{\text{influence}}\times f_{\text{structure}}\times f_{\text{temporal}}$$
>
> where each module can be replaced with a task-specific function. This plug-in design makes the framework general and extensible.
>
> ## W2
> We added Appendix N with Scalability Analysis.
>
> (1) TemGX scales with local neighborhoods, not with the full graph:TemGX operates on the  $\delta $-reachable L-hop neighborhood of each target node. The candidate pool is the induced by its L hop neighbor $(E_L(v_t))$.
>
> (2) Scalability test on the Wiki dataset.We varied both the explanation size k and the number of target nodes $|V_T| $( Appendix N Figure 11(a)(b)).As k increases from 10 to 50, total runtime increases from $2.5 \times 10^3$ s to $6.3 \times 10^3$ s. As $ |V_T| $ increases from 100 to 500, total runtime increases from $1.2 \times 10^3$ s to $ 6.2 \times 10^3 $ s.
>
> ## W3
>
> We clarify that λ is not a tunable hyperparameter, but we suggest a derived constant grounded in our $\delta$-reachability framework (see Appendix L).λ is determined directly by the temporal reachability constraint.  Given a temporal influence term $\ e^{-\lambda \Delta t}$ and  $\delta$-reachability, we suggest:$\lambda = \frac{\alpha}{\delta}$, where $ \alpha \in [0.8, 1.2] $ serves as a normalization factor to account for dataset-specific temporal densities.This design ensures that λ automatically scales with δ, maintaining consistent time decay behavior across datasets without manual tuning.

---

> > ### Author Response · Authors · 2025-11-21
> >
> > ## Q2
> >
> > We added Appendix M (Window Size Sensitivity Analysis) to report how the temporal window size $\delta$ affects TemGX.
> >
> > (1) Effect on UCIM:The results show a inverted-U trend.Small windows ($\delta\$ < 12) miss part of the temporal events needed for the prediction. Fidelity drops to 0.285 at $\delta\$ = 4 (−39%). The best performance appears at $\delta\$ = 12, where the window covers the full temporal range required by the TGNN.When the window becomes too large ($\delta\$ > 12), additional older events enter the candidate pool and introduce temporal noise, reducing fidelity to 0.412 at $\delta\$= 20 (−12%).
> >
> > (2) Effect on Wiki: The results remain steady in the range $\delta\$ in [12, 16], reflecting its denser temporal structure and more uniform event distribution. This indicates that the model already captures the full temporal range, and further expansion offers marginal gains.
> >
> > We suggest choosing $\delta\$ based on domain-specific temporal dependency scales. This ensures robust fidelity while maintaining computational efficiency and temporal interpretability.
> >
> > ## Q3
> >
> > We refer the reviewer to Appendix F and Appendix G for the ablation results of each component across  different tasks.
> >
> > (1) Regression tasks (METR-LA): Removing time decay produces the largest fidelity drop (−16.6%). This indicates that the TGNN prediction depends mainly on recent measurements, so weighting recent events is essential for matching the model’s behavior.
> >
> > (2) Link-prediction tasks (UCIM): Removing time decay results in a smaller fidelity reduction (−14.5%) than other components(−17.1% for TRD and −34.2% for ICM). These tasks rely more on structural and relational influence patterns propagated through L-hop neighborhoods than on temporal recency.
> >
> > Overall, the contribution of time decay is determined by the temporal dependency structure of the task: it plays the leading role when predictions rely heavily on recent information and a smaller role when structural influence dominates.

---

### Official Review · Reviewer_4jHd · 2025-11-01

**Soundness:** 3
**Presentation:** 3
**Contribution:** 3
**Rating:** 8
**Confidence:** 4

**Summary:**

The paper proposes a temporal graph explainer that identifies temporal subgraphs and their solutions based on counterfactual reasoning without any additional training over the base model. The explanation task is framed as a constrained optimization problem over temporal edges and nodes. The detailed experiments show the superiority of the proposed model.

**Strengths:**

- Clear, well-written introduction that makes the problem easy to understand.
- Extensive experiments across multiple datasets with strong baseline comparisons.
- Tackles a difficult problem with a careful, principled formulation.
- Supported by theoretical analysis.
- Open-source code is provided and well documented.

**Weaknesses:**

- The role and sources of randomness in the method are underexplained.
- The number of experimental trials (k = 5) is small for robust statistical conclusions.

**Questions:**

- You define temporal counterfactual edges, and then aggregate corresponding nodes/edges into a temporal subgraph as the explanation. Under what conditions does this subgraph remain a counterfactual? Please clarify your assumptions.
- Given the small number of trials, can you include mean ± standard deviation (and optionally statistical confidence) for all metrics?
- Can you clarify where does stochasticity enter the pipeline?

---

> ### Author Response · Authors · 2025-11-21
>
> We thank the reviewer for the thoughtful feedback and the positive assessment of our work. We address each question below.
> ## W1
>
> We clarify that TemGX is deterministic once the TGNN model is fixed. Our greedy node selection step (Algorithm 1, Line 7) always picks the highest-scoring candidate. The computations of ICM, TRD, and time decay are deterministic when embeddings are fixed.Randomness in our experiments comes only from TGNN training and data splitting, not from the TemGX explanation procedure.
>
> ## W2
>
> We have updated Table 1 to report mean ± standard deviation.
>
> ## Q1
>
> A temporal subgraph $G_\epsilon = (V_s \cup V_c, E_\epsilon) $ is counterfactual if and only if: $M(G_t, v_t) \neq M(G_t \setminus G_\epsilon, v_t) $.
>
> This condition is not automatically satisfied even if individual edges appear counterfactual, because:Removing multiple edges can cause canceling effects, and  alternative temporal paths may carry redundant information.TemGX enforces this through verification at Algorithm 1, Line 8 (shown below):
>
> $$
> \texttt{if Verify}(\mathcal{M}, \mathcal{M}(G_t, v), V_s, v) = \texttt{false then }
> \mathcal{C} := \mathcal{C} \setminus \{v_s^*\}; \texttt{ continue;}
> $$
>
> The verification procedure invokes the TGNN inference twice:(1) Full graph inference:$\hat{y} = M(G_t, v_t) $. (2) Counterfactual graph inference:$\hat{y}' = M(G_t \setminus G_\epsilon, v_t) $ then verification:check whether $ \hat{y} \neq \hat{y}'$.Only nodes and edges that pass this counterfactual verify are retained as explanatory.  If verification fails, TemGX will remove the candidate from the subgraph.
>
> ## Q2
>
> We thank the reviewer for the suggestion and have updated Table 1.
>
> ## Q3
>
> Please refer to our response to W1- TemGX is deterministic.

---

### Official Review · Reviewer_nRh9 · 2025-11-02

**Soundness:** 2
**Presentation:** 2
**Contribution:** 2
**Rating:** 2
**Confidence:** 5

**Summary:**

This paper introduces TemGX, a training-free, model-agnostic framework for explaining predictions from Temporal Graph Neural Networks (TGNNs). The proposed method discovers instance-level temporal subgraphs whose removal changes model predictions, aligning with counterfactual reasoning. TemGX quantifies temporal influence using information cascading and resistance distance metrics, and provides algorithms for efficient explanation generation with guarantees.

**Strengths:**

1 The paper introduces a training-free and queryable framework for generating counterfactual explanations of temporal graph models

2 The proposed method is straightforward and computationally efficient, and it could be adapted for different temporal GNN architectures and tasks.

3 The framework combines temporal influence modeling, resistance distance, and time-decay mechanisms in a coherent formulation supported by an approximation guarantee.

**Weaknesses:**

1 Baseline implementation details are insufficient. It is not mentioned which CoDy variant is used, nor its hyperparameters. Hyperparameters for the other baselines are also not mentioned. Code for the baseline implementations are not provided.

2 Evaluation metrics are defined in an unconventional and problematic way.
Higher sparsity (i.e., larger explanatory subgraphs) is implied to be better (explicitly stated in Appendix E), but this is confusing. Shouldn’t it be that smaller sparsity indicate more concise explanations, and therefore preferred? Instead, the current design seems to reward larger subgraphs, which naturally include more of the L-hop neighborhood and hence boost fidelity and AUFSC scores.
This effect is clearly visible in Figure 4, where the fidelity score of TemGX increases when the sparsity score increases. However, the fidelity score of CoDy basically stopped growing when sparsity reaches around 0.2. Doesn’t this mean that CoDy is superior, since it is able to identify the most decisive but also concise and stable counterfactual explanation?
Moreover, when sparsity equals to 1, it means the entire L-hop neighborhood is used for the explanation, which raises doubts about the usefulness of such explanations. If all available context is included, the explanation is questionable.
In addition, there appears to be inconsistency between Table 1 and Figure 4. While fidelity and AUFSC scores are reported in Table 1, the corresponding sparsity levels are not shown, making it difficult to interpret the trade-off. I would assume that the sparsity scores are fixed for all explanation methods, but this is never clarified. Based on Figure 4 and Table 1, it appears that in some cases, CoDy and TemGX are compared at different sparsity levels. For example, in the UCIM + TGN case, the fidelity scores for TemGX and CoDy are 0.468 and 0.394, respectively, as reported in Table 1. However, Figure 4 shows that TemGX achieves such fidelity score at a sparsity of around 0.4, while CoDy reaches its reported fidelity at a sparsity of around 0.2. This suggests that TemGX’s explanatory subgraph is roughly twice the size of CoDy’s in this setting. Without controlling for sparsity, it becomes unclear whether the fidelity improvement of TemGX truly reflects better explanation quality or simply results from including more of the neighborhood context.
In general, the evaluation is not convincing to me, and I believe this is the biggest weakness of the paper. If this issue can be sufficiently addressed, I would consider to increase the score.

3 Counterfactual analysis is under-specified. No results are provided on the flip-rate (how often the prediction actually changes, or how often counterfactual explanations can be found). Without this, the counterfactual nature of the explanations remains unproven.

4 Classification datasets (Multihost and Elliptic++) are used only in qualitative case studies and are excluded from quantitative results. The paper mentioned these 2 datasets are used for classification, but does not mention the model nor the quantitative explanation results, leaving this part of the evaluation incomplete.

5 Case studies: The case studies are poorly analyzed and not compared with the baselines, and their connection to the claimed strength of TemGX is superficial at most.

6 Poor writing. In general, the writing of the paper should be improved. The example in the introduction is difficult to follow. Figure 1, 2, and 5 all contain many subgraphs that are very small, and not sufficiently described, this is difficult for the readers to follow. The conclusion is overly brief.

7 Missing citation and baseline. It seems that this paper is focusing on discrete time dynamic graphs, but CoDy is designed for continuous time dynamic graphs by its nature. The following paper presents a counterfactual explanation method for DTDGs, it should be included as a baseline, and their evaluation metrics validity and GED should also be taken into consideration.

Prenkaj, B., Villaizán-Vallelado, M., Leemann, T., & Kasneci, G. (2024, August). Unifying Evolution, Explanation, and Discernment: A Generative Approach for Dynamic Graph Counterfactuals. In Proceedings of the 30th ACM SIGKDD Conference on Knowledge Discovery and Data Mining (pp. 2420-2431).

**Questions:**

1 What exact implementation of CoDy and other baselines were used, what are the hyperparameters?

2 Why were Multihost and Elliptic++ not quantitatively evaluated like the others?

3 How does TemGX behave when no counterfactual explanations exist for a prediction?

4 Please address the weaknesses mentioned above.

---

> ### Author Response · Authors · 2025-11-21
>
> ## W1
>
> Thanks! We have clarified the implementation details for all the baseline (see Appendix Section E). We summarize the details below. (1) CoDy variant: We use the CoDy-Spatio-Temporal variant. We found it produced more comprehensive outputs than other variants in the CoDy family. Sample size = 50, candidate size = 128, α = 2/3, neighborhood depth L = 2.
>
> (2) Hyperparameters for the other baselines:
>
> - TemGX:δ = 12, L = 2, k = 50.
> - TGNNExplainer: Trained for 200 epochs with a learning rate of 1e-3 and batch size 32.
> - TempME:Learning rate = 1e-3, contrastive-loss weight = 1.0, dynamic-loss weight = 0.5.
> - Regression datasets (METR-LA, PEMS-BAY): k = 50 (an upper bound of the size of the temporal explanatory subgraphs) is consistently applied for all tests over all these datasets.
>
> (3) Code for all the baselines, including CoDy, TGNNExplainer, and TempME have been provided, in our anonymous github repo (https://anonymous.4open.science/r/TemGX-9D77/README.md) under the folder “baselines/cody”, “baselines/tgnnexp”,”baselines/tempme” respectively.

---

> > ### Author Response · Authors · 2025-11-21
> >
> > ## W2
> >
> > Thanks for the detailed, constructive comments! We address each below.
> >
> > (1) Unconventional metrics. We have followed the convention of GNN explanation literature for the following metrics: For sparsity, we followed the original definition in (Xia et al., “Explorer–Navigator”, 2022): sparsity = $|E_\epsilon| / |E_L(v_t)|$ ,where \$E_\epsilon\$ is the edge set of the explanation subgraph, normalized by $E_L(v_t)$, the edges induced by the L-hop neighbors of the output node $v_t.$ TemGX formulates a justified temporal counterpart $G_ε = (V_s ∪ V_c, E_\epsilon)$ for TGNNs, and accordingly, derives sparsity as  $|E_\epsilon| / |E_L(v_t)|$, where $E_\epsilon\$ refers to the temporal edges among explanatory nodes $ V_s $ as well as the edges connecting them to $v_t$, to ensure the temporal cohesiveness.
> > (2) We remark next, that sparsity is not a “metric” or controlled factor, but are derived from the generated explanations; yet we have necessarily introduced a user-defined size bound k, to allow TemGX discover meaningful explanations as long as they satisfy the size bound. Figure 4 hence is not to evaluate the impact of “sparsity”, but rather an illustration in which we investigate the relationship between the induced sparsity and fidelity. With this, we agree that they reveal, over the tested dataset, that larger temporal subgraphs have higher fidelity. We consider this happens to justify our design that will not discourage the discovery of larger temporal subgraphs, as: (i) in a temporal setting, it may take more number of temporal edges to ensure temporal cohesiveness in terms of $\delta$-reachability; and (ii) while conciseness remains to be a good measure, we advocate the importance of large, informative explanatory structures that provide a trace of “evolving” structures; hence our setting does not prevent us to discover such structures for temporal GNNs. We also clarify that our design is not necessarily “rewarding” large structures, as one can easily verify that our objective function is not monotonic to $|E_\epsilon|$: a larger, or smaller structure does not necessarily indicate a higher score. The optimal structure is only constrained by the size bound k. The actual size and quality is dataset-dependent. Hence we remark that the trend we observe in Figure 4 is test and dataset dependent, but not a necessary consequence of applying our design.
> >
> > On the other hand, to further clarify your confusion, we have included the following analysis in the revised manuscript. (1) “Why CoDy converge early in Figure 4?” CoDy’s fidelity curve coverage near sparsity ≈ 0.2. We found that CoDy’s MCTS (with α = 2/3 as recommended in (Qu et al., ICML 2025 Appendix G)) encourages early-discovery of decisive events, hence tends to find smaller perturbations that flip predictions and yield locally decisive explanation structures. While k is larger, the remaining temporal edges, some may still form $\delta$-reachability and higher temporal influence and from relatively older history of users’ interests, remains not explored. As a result, fidelity is improved faster, yielding locally decisive explanations.
> >
> > (2) “Why TemGX could improve fidelity as k increases?” TemGX ranks events with ICM × TRD score, that combines influence propagation and temporal–structural connectivity. As k becomes larger, temporal explanations including “more” such edges (hence “larger”) may have a chance to further improve fidelity – even our optimization goal has no intention to encourage larger structures – because they indeed introduce more predictive influence. TemGX does not prevent us to discover such larger structures, as long as they have bounded size k and ensure temporal cohesiveness under $\delta $-reachability. (3) We have also added Figure 9  to evaluate the impact of k, which verifies that a larger upper bound may help us discover explanations with higher fidelity – yet these only mean more explanation structures that improve the fidelity are allowed to be found, not necessarily mean they are larger.
> >
> > We conclude by considering that TemGX and Cody are complementary, rather than competing solutions that one must be “better” than the other. Cody benefits in a case of “think fast” and satisfy users’ need for quickly generating concise, more local explanations, while TemGX may be suitable for “think slower” yet allow the discovery of explanations that span longer in history.（Liu, Chen, Liu, Zhang & Xie,Multi-objective explanations of GNN predictions, 2021), (Kahneman, Thinking, Fast and Slow, 2011) Our case study (Table 7) provides application scenarios that both Cody and TemGX may benefit, in complementary ways of contributing small (not necessarily temporally connected) and larger, temporally cohesive counterparts from TemGX.

---

> ### Author Response · Authors · 2025-11-21
>
> ## W2
> (4) Sparsity, in theory, may be 1; yet this case rarely shows up in our tests. TemGX will stop selecting edges once $\delta $-reachable temporal structure is satisfied or the score Φ(E_s) no longer increases (see Algorithm TemGX line:9-14); and In counterfactual verification, edges that do not affect the prediction are rejected. Indeed, across all datasets (Figure 4), TemGX coverage around sparsity ≈ 0.4–0.6 < 1.
>
> ## W3
>
> We have added flip-rate analysis in Appendix K Table 8. For a target node $v_t $ with prediction $M(G_t, v_t) = y_{\text{pred}} \$, we define $\text{Flip-rate} = \frac{|\{ v_t \in V_T : M(G_t \setminus G_{\epsilon}, v_t) \neq y_{\text{pred}} \}|}{|V_T|}\$,where $ G_{\epsilon} $ is the explanatory subgraph. For classification, “$\neq\$” means a different predicted class. For regression, “$\neq\$” means $\(|M(G_t) - M(G_t \setminus G_{\epsilon})| > 0.05 \times |M(G_t)| \$. This follows the standard notion of prediction flips used in counterfactual GNN explanations (Lucic et al., 2022, Section 4). Our test shows that TemGX can effectively identify temporal structures with size bound k = 50 with Flip-rates largely between 80–90%. We expect a flip-rate approaching 100% with larger k, and the entire temporal graph G is a solution if no size bound is posed k=$\infty\$ by definition.
>
> ## W4 and W5
>
> We have included a new Table 7 in Appendix H. Using Wiki dataset, we illustrate a visual comparison of the explanations produced by all baselines and TemGX. These visualizations make the differences between methods clear and show how TemGX forms $\delta\$-reachable and temporally coherent structures that follow the temporal patterns in the data.
>
> Both Multihost and Elliptic++ are node-classification datasets that provide node-level labels. The baseline explainers CoDy, TGNNExplainer, and TempME are designed for link prediction and require edge-level supervision. These datasets do not include the edge-level ground truth needed by the baselines, so a quantitative comparison cannot be carried out in a consistent setting.
>
> We have added Appendix K Table 8, where we report the average fidelity of TemGX on both datasets. This addition shows how TemGX behaves on node-classification tasks. TemGX can work on link prediction, node classification, and regression, so it can be applied across all datasets used in the paper.
>
> Baseline compatibility:
>
> | Baseline      | Designed Task          | Compatible Datasets                                  |
> | ------------- | ---------------------- | ---------------------------------------------------- |
> | CoDy          | Link Prediction        | UCIM, Wiki                                           |
> | TGNNExplainer | Link Prediction        | UCIM, Wiki                                           |
> | TempME        | Link Prediction        | UCIM, Wiki                                           |
> | TemGX         | Link, Node, Regression | UCIM, Wiki, Multihost, Elliptic++, METR-LA, PEMS-BAY |
> ## W6
>
> We have implemented the requested improvements increasing figure font sizes.
>
> ## W7
>
> GRACIE addresses counterfactual explanations in discrete-time dynamic graphs (DTDGs), where graph evolution is modeled as a sequence of snapshots and explanations are generated at the snapshot level. TemGX operates in the continuous-time dynamic graph (CTDG) setting formalized by CoDy (Qu et al., ICML 2025), where interactions occur at timestamp events and temporal aggregation depends on temporal distances. For this reason, GRACIE is included as related work but addresses a different problem and is not a directly comparable baseline for the CTDG setting considered by TemGX.
> ## Q1
>
> Please refer to our response to W1.
>
> ## Q2
>
> Please refer to our response to W4 and W5.
>
> ## Q3
> When no counterfactual explanation exists under a size bound k and requirement of $\delta\$-reachability, TemGX will performs the full search procedure in Figure 3, including the replacement policy and the connecting-node expansion (lines 5–14), attempting to assemble any $\delta\$-reachable temporal candidates that could affect the prediction.However, when all candidates—including directly influential nodes and connecting nodes—fail the Verify(·) check, the objective $E_s$ never increases and the loop naturally terminates and return an empty set.

---

### Author Response · Authors · 2025-12-03
**Author Response Summary to the Area Chair**

We thank all reviewers for their careful reading and constructive feedback. Our rebuttal addresses each concern in detail, and we have incorporated all necessary clarifications and additions into the revised manuscript. All newly added or revised text is highlighted in red across Sections 1, 3, 5, and 6, as well as Appendix E and H, and the newly added Appendix J–N. Below, we summarize our responses for each reviewer.

For reviewer nRh9, we provided a detailed response to all concerns. In particular, we expanded the analysis requested in W2, including (1) a principled explanation of the sparsity–fidelity relationship following standard temporal GNN explanation formulations, (2) analyses clarifying why TemGX’s fidelity increases with k, (3) the role of $\delta$-reachability and temporal cohesiveness, and (4) an additional figure evaluating fidelity under fixed size bounds (Appendix J). These directly address the reviewer’s main concern, which was identified as “the biggest weakness of the paper,” and the reviewer noted that resolving this issue could lead to an increased score. We also clarified baseline implementations (Appendix E), added flip-rate analysis (Appendix K), included fidelity results for Multihost and Elliptic++ (W4–W5), increased figure font sizes (W6), incorporated GRACIE into the related work (W7), and explained the behavior of TemGX when no counterfactual exists (Q3).

For reviewer 4jHd, who raised questions regarding the determinism of TemGX, the precise conditions under which a temporal subgraph remains counterfactual, and the need for mean ± standard deviation, we clarified that TemGX is fully deterministic once the TGNN is fixed, formalized the counterfactual validity condition in Algorithm 1, and added mean ± std for all metrics (updated Table 1).

For reviewer PEm2, who raised concerns about (W1) the novelty of our temporal explainability measure, (W2) scalability, (W3/Q2) the sensitivity to $\delta$, and (Q3) the role of time decay, we expanded the manuscript with targeted clarifications. In Section 3 and Appendix F–G, we clearly formalize our multiplicative Φ score, define Temporal RD over $\delta$-reachable time, prove submodularity, and show through ablations that each component contributes differently across tasks. In Appendix N, we added a scalability study demonstrating that runtime grows with the local $\delta$-reachable neighborhood rather than full graph size. In Appendix L, we explain that λ is derived from δ (λ≈α/δ), avoiding hyperparameter sweeping. In Appendix M, we provide window-size sensitivity analysis with consistent trends across datasets. We also clarify in Appendix F–G why time decay provides the largest gains in regression tasks but smaller gains in link prediction.

For reviewer Kj7f, who asked how our work addresses the limitations raised in the prior paper ( "On the Limit of Explaining Black-box Temporal Graph Neural Networks" (https://arxiv.org/pdf/2212.00952) )  regarding perturbation-based temporal GNN explainers, we clarified this connection by summarizing how TemGX addresses the three limits identified in that work through δ-reachability, the multiplicative influential score, and counterfactual verification. These mechanisms ensure that TemGX does not rely on perturbation alone and therefore avoids the limitations discussed in the cited paper.

---

### Meta-Review · Area_Chair_XcXP · 2026-01-04

**Summary:**

This paper introduces TemGX, a training-free, post-hoc framework for explaining predictions of Temporal Graph Neural Networks (TGNNs). TemGX discovers temporal subgraphs responsible for model outputs, leveraging a multiplicative influence score that integrates temporal influence propagation (ICM), temporal resistance distance (TRD), and time decay. The framework is queryable, efficient, and comes with approximation guarantees. Extensive experiments demonstrate TemGX’s effectiveness compared to state-of-the-art explainers.

The rebuttal was thorough and convincing, addressing nearly all reviewer concerns with new experiments, clarifications, and expanded analyses. While Reviewer nRh9 had a reject rating, the additional fidelity–sparsity analysis and flip-rate results strengthen the case for TemGX. I beleive there is a reasonable chance that this reviewer would increase the rating had the full discussion period remained intact. Other reviewers (4jHd, PEm2, Kj7f) are all positive.

Overall, the novelty, technical soundness, and rebuttal responsiveness justify acceptance.

**Reviewer Concerns:**

The rebuttal is thorough and I believe all concerns were reasonably addressed. The key concerns addressed includes the follows:

* Baseline implementation details and hyperparameters clarified, with code provided (it appears the link has expired, the authors should fix this).

* Sparsity–fidelity relationship explained, with added analyses and figures.

* Flip-rate analysis included, demonstrating counterfactual validity.

* Novelty of temporal explainability measure justified via ablation studies.

* Scalability analysis

* Connection to prior work (Vu & Thai, 2022) explicitly addressed.

Finally, I think some skepticism persists regarding the evaluation design, especially fidelity gains from larger subgraphs, but this does not hinder acceptance given the paper’s overall contribution.

**Reviewer Scores:**

Two reviewers had ratings of 8, which are likely to remain unchanged. Another reviewer rated the paper a 6. The fourth reviewer was most critical with a rating of 2, but given the strong and comprehensive rebuttal, it is reasonable to expect this score to rise to 6.

---

### Decision · Program_Chairs · 2026-01-26

Accept (Poster)